# Analysis of ultrasonic vocalizations from mice using computer vision and machine learning

Antonio HO Fonseca[1,2†]*, Gustavo M Santana[1,2,3†]*, Gabriela M Bosque Ortiz[1,4]*, Sérgio Bampi[2]*, Marcelo O Dietrich[1,3,4,5]*

[1]Laboratory of Physiology of Behavior, Department of Comparative Medicine, Yale School of Medicine, New Haven, United States; [2]Institute of Informatics, Federal University of Rio Grande do Sul, Porto Alegre, Brazil; [3]Graduate Program in Biological Sciences - Biochemistry, Federal University of Rio Grande do Sul, Porto Alegre, Brazil; [4]Interdepartmental Neuroscience Program, Biological and Biomedical Sciences Program, Graduate School in Arts and Sciences, Yale University, New Haven, United States; [5]Department of Neuroscience, Yale School of Medicine, Porto Alegre, Brazil

*For correspondence:
antonio.fonseca@yale.edu (AHOF);
gustavo.santana@yale.edu (GMS);
gabriela.borque@yale.edu (GMBO);
bampi@inf.ufrgs.br (SB);
marcelo.dietrich@yale.edu (MOD)

Present address:
[†]Interdepartmental Neuroscience Program, Yale School of Medicine, New Haven, United States

Competing interests: The authors declare that no competing interests exist.

**Abstract** Mice emit ultrasonic vocalizations (USVs) that communicate socially relevant information. To detect and classify these USVs, here we describe VocalMat. VocalMat is a software that uses image-processing and differential geometry approaches to detect USVs in audio files, eliminating the need for user-defined parameters. VocalMat also uses computational vision and machine learning methods to classify USVs into distinct categories. In a data set of >4000 USVs emitted by mice, VocalMat detected over 98% of manually labeled USVs and accurately classified ≈86% of the USVs out of 11 USV categories. We then used dimensionality reduction tools to analyze the probability distribution of USV classification among different experimental groups, providing a robust method to quantify and qualify the vocal repertoire of mice. Thus, VocalMat makes it possible to perform automated, accurate, and quantitative analysis of USVs without the need for user inputs, opening the opportunity for detailed and high-throughput analysis of this behavior.

## Introduction

In animals, vocal communication transmits information about the state of the caller and influences the state of the listener. This information can be relevant for the identification of individuals or groups (*Hoffmann et al., 2012*); status within the group (e.g., dominance, submission, fear, or aggression; *Nyby et al., 1976*); next likely behavior (e.g., approach, flee, play, or mount; *Neunuebel et al., 2015*); environmental conditions (e.g., presence of predators, location of food; *Slobodchikoff et al., 2012*); and facilitation of mother–offspring interactions (*D'Amato et al., 2005*).

Mice emit ultrasonic vocalizations (USVs) in a frequency range (≈30–110 kHz) above the human hearing range (≈2–20 kHz) (*Zippelius and Schleidt, 1956*; *Noirot, 1972*; *Nyby et al., 1976*; *Nyby et al., 1977b*; *Nyby et al., 1977a*; *Sales and Smith, 1978*; *Branchi et al., 2001*; *Hahn and Lavooy, 2005*; *Ehret, 2005*; *Branchi et al., 2006*). These USVs are organized in *phrases* or *bouts* composed of sequences of *syllables*. The syllables are defined as continuous units of sound not interrupted by a period of silence. The syllables are composed of one or more notes and are separated by salient pauses and occur as part of sequences (*Arriaga et al., 2012*; *Holy and Guo, 2005*). These transitions across syllables do not occur randomly (*Holy and Guo, 2005*; *Castellucci et al., 2018*),

and the changes in the sequences, prevalence, and acoustic structure of syllables match current behavior (*Chabout et al., 2015*), genetic strain (*Van Segbroeck et al., 2017*; *Scattoni et al., 2011*), and developmental stage (*Grimsley et al., 2011*). USVs are commonly emitted by mouse pups when separated from the home nest (*Scattoni et al., 2008*) and are modulated during development (*Grimsley et al., 2011*; *Elwood and Keeling, 1982*; *Castellucci et al., 2018*). In the adult mouse, USVs are emitted in both positive and negative contexts (*Arriaga and Jarvis, 2013*). Thus, understanding the complex structure of USVs will advance vocal and social communications research.

In the past years, tools for USV analysis advanced significantly (*Coffey et al., 2019*; *Van Segbroeck et al., 2017*; *Neunuebel et al., 2015*; *Chabout et al., 2015*; *Arriaga et al., 2012*; *Tachibana et al., 2020*). For the detection of USVs in audio recordings, the majority of the software tools available depend on user inputs (*Neunuebel et al., 2015*; *Van Segbroeck et al., 2017*; *Tachibana et al., 2020*) or present limited detection capabilities (*Arriaga et al., 2012*; *Chabout et al., 2015*). An exception is DeepSqueak (*Coffey et al., 2019*), which uses a neural network-based method for the detection of USVs from audio recordings. For the classification of USVs, different tools use supervised classification (*Arriaga et al., 2012*; *Chabout et al., 2015*; *Coffey et al., 2019*) and unsupervised clustering (*Burkett et al., 2015*; *Van Segbroeck et al., 2017*; *Coffey et al., 2019*) methods to assign USVs to different groups. However, no consensus exists on the biological function of the various USV subclasses, making it challenging to develop a tool for all purposes. Additionally, the accuracy of classification methods depends on the accurate detection of vocalizations in audio recordings, which remains a challenge in experimental conditions that include noise and low intensity vocalization. To overcome these limitations, our goal was to create a tool that does not need user inputs to detect and classify USVs with high accuracy and that extracts the spectral features of the USVs with precision.

Here, we describe the development of VocalMat, a software for robust and automated detection and classification of mouse USVs from audio recordings. VocalMat uses image processing and differential geometry approaches to detect USVs in spectrograms, eliminating the need for parameter tuning. VocalMat shows high accuracy in detecting USVs in manually validated audio recordings, preserving quantitative measures of their spectral features. This high accuracy allows the use of multiple tools for USV classification. In the current version of VocalMat, we embedded a supervised classification method that uses computer vision techniques and machine learning to label each USV into 11 classes. Thus, VocalMat is a highly accurate software to detect and classify mouse USVs in an automated and flexible manner.

## Results

### Detection of mouse USVs using imaging processing

VocalMat uses multiple steps to analyze USVs in audio files (see *Figure 1A* for the general workflow). Initially, the audio recordings are converted into high-resolution spectrograms through a short-time Fourier transformation (see Materials and methods). The resulting spectrogram consists of a matrix, wherein each element corresponds to an intensity value (power spectrum represented in decibels) for each time-frequency component. The spectrogram is then analyzed as a gray-scale image, where high-intensity values are represented by brighter pixels and low-intensity values by darker pixels (*Figure 1B*). The gray-scale image undergoes contrast enhancement and adaptive thresholding for binarization (see Materials and methods). The segmented objects are further refined via morphological operations (*Figure 1C* and *Figure 1—figure supplement 1*), thus resulting in a list of segmented blobs (hereafter referred to as USV candidates) with their corresponding spectral features (*Figure 1D*). Finally, because experimental observations demonstrate a minimum of 10 ms of interval between two successive and distinct USVs, we combined into a single USV candidate blobs that were separated for less than 10 ms. The final list of USV candidates contain real USVs and noise (i.e., detected particles that are not part of any USV).

To reduce the amount of data stored for each USV, the features extracted from detected candidates are represented by a mean frequency and intensity every 0.5 ms. The means are calculated for all the individual candidates, including the ones overlapping in time, hence preserving relevant features such as duration, frequency, intensity, and harmonic components (*Figure 1D*).

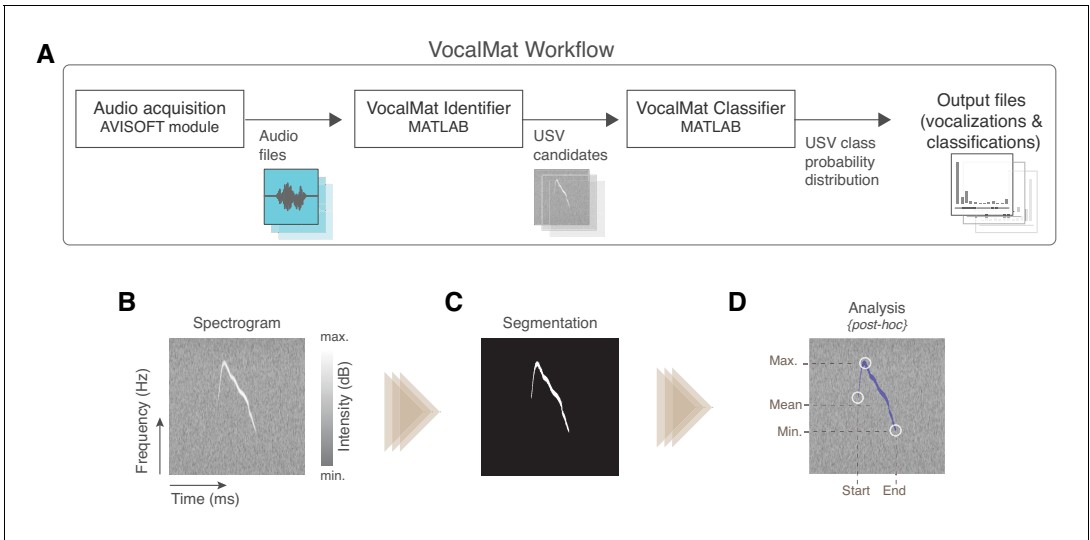

**Figure 1.** Overview of the VocalMat pipeline for ultrasonic vocalization (USV) detection and analysis. (A) Workflow of the main steps used by VocalMat, from audio acquisition to data analysis. (B) Illustration of a segment of spectrogram. The time-frequency plan is depicted as a gray scale image wherein the pixel values correspond to intensity in decibels. (C) Example of segmented USV after contrast enhancement, adaptive thresholding, and morphological operations (see *Figure 1—figure supplement 1* for further details of the segmentation process). (D) Illustration of some of the spectral information obtained from the segmentation. Information on intensity is kept for each time-frequency point along the segmented USV candidate. The online version of this article includes the following figure supplement(s) for figure 1:

**Figure supplement 1.** Image processing pipeline for segmentation of ultrasonic vocalizations (USVs) in spectrograms.

Harmonic components are also referred to as nonlinear components or composite (*Scattoni et al., 2008*). Here, we did not consider harmonic components as a different USV, but rather as an extra feature of a USV (*Grimsley et al., 2011*). Therefore, each detected USV candidate may or may not present a harmonic component. A harmonic component was considered as a continuous sound (i.e., no discontinuities in time and/or frequency) overlapping in time with the main component of the USV (similar to *Grimsley et al., 2011*).

Besides the list of USV candidates and their spectral features, the segmentation process also exports image files of 227 × 227 pixels, in which the USV candidate is centralized in windows of 220 ms (see *Figure 1B*). This temporal length is defined as twice the maximum duration of USVs observed in mice (*Grimsley et al., 2011*), thus preventing cropping.

## Eliminating noise using a contrast filter

Initially, we used VocalMat to detect USVs in a set of 64 audio recordings. These recordings were composed of experiments using mice of different strains, age, sex, and in a variety of experimental conditions (e.g., recorded inside a chamber that produce high levels of environmental noise) to increase the variability of the data set. In this data set, VocalMat initially detected a pool of 59,781 USV candidates, which includes real USVs and noise (*Figure 2A* and Materials and methods). Visual inspection of the data set revealed that artifacts generated during the segmentation process dominated the pool of USV candidates (see *Figure 2B* for examples of real USVs and noise in the pool of USV candidates). This type of artifact is characterized by its low intensity compared to real USVs. To remove these artifacts from the pool of USV candidates, we applied a *Local Median Filter* step, a method to estimate the minimum expected contrast between a USV and its background for each audio recording. This contrast is calculated based on the median intensity of the pixels in each detected USV candidate $k$ (referred to as $\widehat{X_k}$), and the median intensity of the background pixels in a bounding box containing the candidate $k$ (referred to as $\widehat{W_k}$) (*Figure 2C*). Thus, the contrast is defined as the ratio $C_k = \widehat{X_k}/\widehat{W_k}$.

To validate this method, a group of investigators trained to detect USVs manually inspected the spectrograms and labeled the USVs in a subset of seven randomly selected audio recordings (hereafter referred to as *test* data set and described in Table 3). Each USV was labeled by at least two

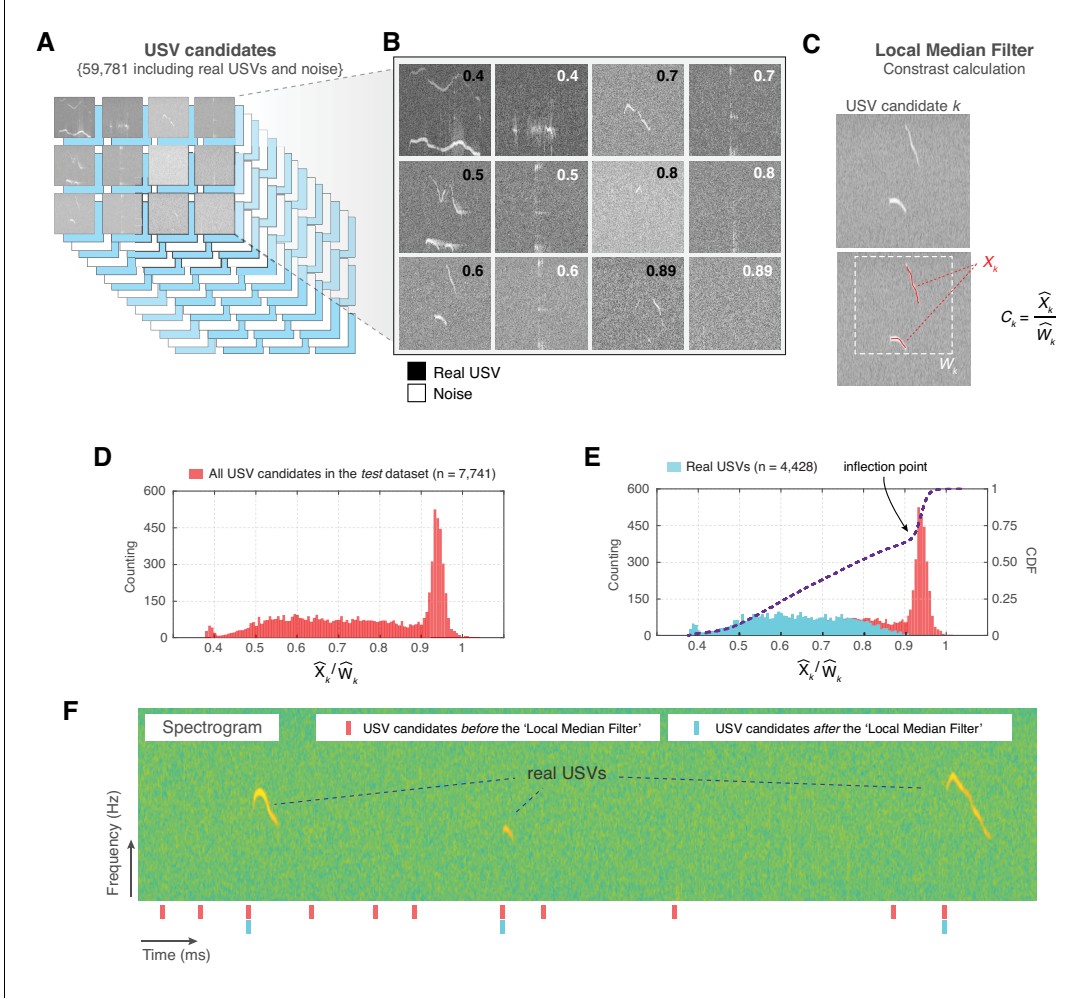

**Figure 2.** Noise elimination process for ultrasonic vocalization (USV) candidates. (**A**) In a set of 64 audio files, VocalMat identified 59,781 USV candidates. (**B**) Examples of USVs among the pool of candidates that were manually labeled as either noise or real USVs. The score (upper-right corner) indicates the calculated contrast $C_k$ for the candidate. (**C**) Example of contrast calculation ( $C_k$) for a given USV candidate $k$. The red dots indicate the points detected as part of the USV candidate ($X_k$) and the dashed-white rectangle indicates its evaluated neighborhood ($W_k$). (**D**) Distribution of the $C_k$ for the USV candidates in the *test* data set. (**E**) Each USV candidate was manually labeled as real USV or noise. The distribution of $C_k$ for the real USVs (cyan) compared to the distribution for all the USV candidates (red) in the *test* data set. The blue line indicates the cumulative distribution function (CDF) of $C_k$ for all the USV candidates. The inflection point of the CDF curve is indicated by the arrow. (**F**) Example of a segment of spectrogram with three USVs. The analysis of this segment without the 'Local Median Filter' results in an elevated number of false positives (noise detected as USV). 'Red' and 'cyan' ticks denote the time stamp of the identified USV candidates without and with the 'Local Median Filter', respectively.

investigators and when discrepancy occurred, both individuals reviewed their annotation under the supervision of a third investigator. (This data set contains fully validated audio recordings with manually inspected USVs and is publicly available to facilitate the development and the test of performance of new tools.)

In the *test* data set, a total of 7741 USV candidates were detected using the segmentation process described above, representing 1.75 times more USV candidates than the manual counting (4441 USVs). Importantly, the segmentation step included 4428 real USVs within the pool of USV candidates, therefore missing 13 USVs compared to the ground-truth.

The distribution of $C_k$ for real USVs and for noise showed that the peak at high $C_k$ (i.e., low contrast) in the distribution was dominated by USV candidates corresponding to noise (*Figure 2D,E*). The $C_k$ of real USVs (mean = 0.642, SEM = 1.841 $\times 10^{-3}$, median = 0.640, 95% CI [0.638, 0.646]; N = 4,428) was significantly lower than the $C_k$ of noise (mean = 0.922, SEM = 9.605 $\times 10^{-4}$, median = 0.936, 95% CI [0.921, 0.924]; n = 3,336; p < $10^{-15}$, D = 0.894, Kolmogorov–Smirnov test;

*Figure 2D,E*). This unbalanced bimodal distribution causes an inflection point on the cumulative distribution function (CDF) of $C_k$ that matches the ratio observed for segmentation artifacts (*Figure 2E*). Therefore, based on these results, we used the automatically calculated inflection point, which is specific to each audio recording, as a threshold to effectively eliminate a substantial amount of noise from the pool of USV candidates (details on this calculation are provided in Materials and methods).

In the *test* data set, 5171 out of 7741 USV candidates survived the *Local Median Filter* step. This number includes real USVs (4421; out of the 4428 real USVs after automatic segmentation) and remaining noises of lower $C_k$. Thus, this step eliminated seven real USVs of the pool of candidates, all of which presented a high $C_k$ (mean = 0.942, SEM = 5.871 $\times 10^{-3}$, median = 0.943, 95% CI [0.927, 0.956]; n = 7). The remaining noises among the pool of candidates had high intensity and were commonly originated from external sources (*Figure 2B,E*).

To illustrate the performance of the *Local Median Filter* step, *Figure 2F* shows a segment of a spectrogram with 11 USV candidates detected and three real USVs. After applying the *Local Median Filter* step, only the real USVs remained in the pool of USV candidates. Thus, the *Local Median Filter* step effectively eliminates segmentation noise from the pool of USV candidates, which provides two main advantages: first, it decreases the number of USV candidates used in downstream analysis and, second, it reduces the number of false positives.

In an ideal experimental setting with complete sound insulation and without the generation of noise by the movement of the animal, no further step is required to detect USVs using VocalMat. Since this is difficult in experimental conditions, we applied a second step in the noise elimination process.

## Using convolutional neural network for noise identification

To identify USVs in the pool of USV candidates that passed the *Local Median Filter* step, we trained a convolutional neural network (CNN) to classify each USV candidate into one of 11 USV categories or noise (see Figure 4A for examples of the different USV categories). We used a data set containing 10,871 samples manually labeled as one of the 11 USV categories and 2083 samples manually labeled as noise (see Materials and methods). The output of the CNN is the probability of each USV candidate belonging to one of the 12 categories. The most likely category defines the label of the USV candidate (*Figure 3A*).

To evaluate the performance of VocalMat in distinguishing between USVs and noise, we used the 5171 USV candidates in the *test* data set that passed the *Local Median Filter* step (Materials and methods). We compared the probability for the label *Noise* ($P(Noise)$) to the sum over the probabilities of the 11 USV categories ($P(USV)$). The rate of detected USVs labeled as such (true positive or sensitivity) was 99.04 ± 0.31% (mean ± SEM; median = 99.37; 95% CI [98.27, 99.80]). The rate of detected USVs labeled as noise (false negative) was 0.96 ± 0.31% (mean ± SEM; median = 0.61; 95% CI [0.20, 1.73]). The rate of detected noise labeled as noise (true negative rate or specificity) was 94.40 ± 1.37% (mean ± SEM; median = 95.60; 95% CI [91.60, 97.74]). The rate of detected noise labeled as USV (false positive) was 5.60 ± 1.37% (mean ± SEM; median = 4.40; 95% CI [2.26, 8.94]), representing a total of 42 wrongly detected USVs out of the 5171 USV candidates in the *test* data set. Altogether, the accuracy in identifying USVs was 98.63 ± 0.20% (mean ± SEM; median = 98.55; 95% CI [98.14, 99.11]) for manually validated audio recordings. (The performance of VocalMat in identifying USVs in individual audio recordings is provided in *Table 1*.) Thus, Vocal-Mat presents high accuracy to detect USVs from audio recordings and to remove noise (*Figure 3B*), failing to identify approximately 1 in 75 USVs.

We further calculated other measures of performance (*Figure 3C*). For USVs wrongly labeled as noise (false negative), the probability of being a USV was 0.15 ± 0.03 (mean ± SEM; median = 0.04; 95% CI [0.09, 0.22]; *Figure 3C*), while for noise labeled as USV (false positive), the probability of being USV was 0.85 ± 0.03 (mean ± SEM; median = 0.86; 95% CI [0.80, 0.91]; *Figure 3C*). These probabilities contrast with cases in which VocalMat correctly identified USV and noise. USVs that were correctly identified had a probability of being USV of 0.99 ± 3.78 $\times 10^{-4}$ (mean ± SEM; median = 1.00; 95% CI [0.99, 0.99]; *Figure 3C*). Noise that was correctly identified had a probability of being noise of 0.99 ± 1.78 $\times 10^{-3}$ (mean ± SEM; median = 1.00; 95% CI [0.98, 0.99]; *Figure 3C*). These results indicate that the probability assigned by VocalMat *flags* likely errors in classification.

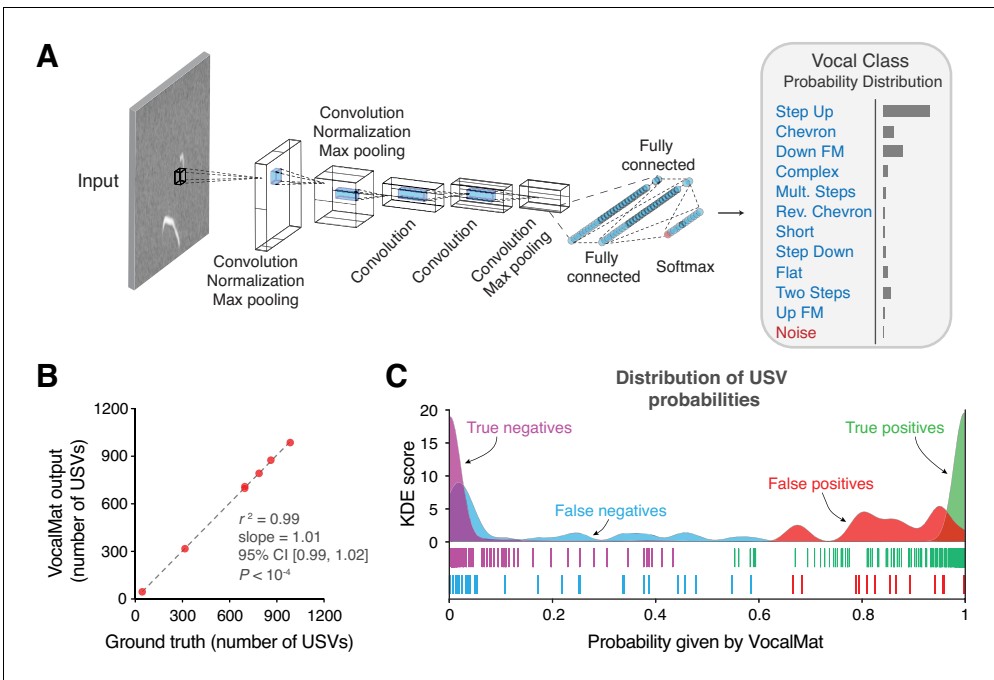

**Figure 3.** VocalMat ultrasonic vocalization (USV) classification using a convolutional neural network. (**A**) Illustration of the AlexNet architecture post end-to-end training on our *training* data set. The last three layers of the network were replaced in order to perform a 12-categories (11 USV types plus noise) classification task. The output of the CNN is a probability distribution over the labels for each input image. (**B**) Linear regression between the number of USVs manually detected versus the number reported by VocalMat for the audio files in our *test* data set (see *Figure 4—figure supplement 1* for individual confusion matrices). (**C**) Distribution of probabilities $P(USV)$ for the true positive (green), false positive (red), false negative (cyan), and true negative (magenta). Ticks represent individual USV candidates.

These *flagged* candidates (i.e., with assigned low probability) can be manually inspected to correct the misclassification and retrain VocalMat.

## Performance of VocalMat compared to other tools

Next, we tested the performance of VocalMat in comparison to other software tools. We first searched for validated data sets that were used by previous tools. We obtained 15 audio recordings made publicly available by USVSEG (*Tachibana et al., 2020*) and one audio recording made publicly available by DeepSqueak (*Coffey et al., 2019*).

The USVSEG data set consisted of five audio recordings of 5–6 days old mouse pups containing 409 USVs and 10 audio recordings of adult mice containing 2401 USVs. The accuracy of VocalMat in identifying USVs in the USVSEG data set was 98.62 ± 0.53 (mean ± SEM, median = 99.61, 95% CI

**Table 1.** Summary of performance of VocalMat in detecting ultrasonic vocalizations (USVs) in the *test* data set.

| Audio file | True positive | False negative | True negative | False positive | Accuracy (%) |
|---|---|---|---|---|---|
| 1 | 316 | 1 | 58 | 2 | 99.20 |
| 2 | 985 | 1 | 105 | 15 | 98.55 |
| 3 | 696 | 12 | 73 | 5 | 97.84 |
| 4 | 862 | 13 | 51 | 4 | 98.17 |
| 5 | 44 | 1 | 216 | 3 | 98.48 |
| 6 | 696 | 2 | 87 | 4 | 99.24 |
| 7 | 787 | 5 | 122 | 5 | 98.91 |

[97.49, 99.76]), which is comparable to the approximately 98% accuracy reported by USVSEG (*Tachibana et al., 2020*). However, VocalMat detected approximately 6% more USVs than USVSEG. More precisely, in the audio recordings from pups, VocalMat presented a true positive rate of 97.53 ± 1.04% (mean ± SEM, median = 98.41, 95% CI [94.65, 100.4]), detecting 15 more USVs than USV-SEG. In the audio recordings from adult mice, VocalMat presented a true positive rate of 99.05 ± 0.37% (mean ± SEM, median = 99.39, 95% CI [98.20, 99.90]), detecting 152 more USVs than USVSEG.

Similar to VocalMat, DeepSqueak uses deep learning to detect USVs (*Coffey et al., 2019*). To directly compare DeepSqueak and VocalMat, we evaluated the performance of both tools on the single audio recording provided by DeepSqueak (*Coffey et al., 2019*). First, we manually inspected the spectrogram and labeled each of the 762 USVs identified. Of these 762 USVs, VocalMat detected 747 with a true positive rate of 91.73%, whereas DeepSqueak detected 608, with a true positive rate of 77.95%. Thus, when tested in data sets from different laboratories, VocalMat shows better sensitivity for USV detection than DeepSqueak and USVSEG.

As described above, the *test* data set was fully curated by manual identification of USVs. To take advantage of this large data set, we further compared the performance of VocalMat with four tools (see Materials and methods). In addition to USVSEG (*Tachibana et al., 2020*) and DeepSqueak (*Coffey et al., 2019*), we also tested the performance of Ax (*Neunuebel et al., 2015*) and MUPET (*Van Segbroeck et al., 2017*). (*Table 2* summarizes the performance of these tools in our *test* set.)

Ax requires a series of manual inputs for their detection algorithm (*Neunuebel et al., 2015*). Combining the best configurations tested (*Supplementary file 1*), the percentage of missed USVs was 4.99 ± 1.34% (mean ± SEM; median = 4.07, 95% CI [1.73, 8.26]) and the false discovery rate was 37.67 ± 5.59% (mean ± SEM; median = 42.56, 95% CI [23.99, 51.34]). In comparison to Ax, MUPET has a lower number of parameters to be set by the user. Combining the best configurations tested (*Supplementary file 2*), the percentage of missed USVs was 33.74 ± 3.81% (mean ± SEM; median = 33.13, 95% CI [24.41, 43.07]) and the false discovery rate was 38.78 ± 6.56% (mean ± SEM; median = 32.97, 95% CI [22.72, 54.84]). Similar to Ax and MUPET, USVSEG requires setting parameters manually for USV detection (*Supplementary file 3*). USVSEG displayed the best performance out of the manually configured tools, presenting a missed vocalization rate of 6.53 ± 2.56% (mean ± SEM; median = 4.26, 95% CI [0.26, 12.80]) and a false discovery rate of 7.58 ± 4.31% (mean ± SEM; median = 3.27, 95% CI [−2.97, 18.15]). It is important to emphasize that the tests with Ax, MUPET, and USVSEG did not explore all possible combinations of parameters and, therefore, other settings could potentially optimize the performance of these tools to detect USVs in the *test* data set.

Finally, we compared the performance of DeepSqueak with VocalMat (*Supplementary file 4*). The best values obtained were a rate of missed USVs of 27.13 ± 3.78% (mean ± SEM; median = 24.22, 95% CI [17.86, 36.40]) and a false discovery rate of 7.61 ± 2.35% (mean ± SEM; median = 4.73, 95% CI [1.84, 13.39]). The manual inspection of the USVs detected by DeepSqueak revealed cases of more than one USV being counted as a single USV, which could lead to an inflated number of missed USVs. Since we did not train DeepSqueak with our data set, it is possible that DeepSqueak could present much better performance than what we report here when custom-trained. Thus, using both external data sets from different laboratories and our own fully validated *test* data set, Vocal-Mat presents high performance in detecting USVs in audio recordings without the need for *any* parameter tuning or custom training of the CNN.

**Table 2.** Summary of detection performance.

| Tool | Missed ultrasonic vocalizations (USVs) rate (%) | False discovery rate (%) |
| --- | --- | --- |
| Ax | 4.99 | 37.67 |
| MUPET | 33.74 | 38.78 |
| USVSEG | 6.53 | 7.58 |
| DeepSqueak | 27.13 | 7.61 |
| VocalMat | 1.64 | 0.05 |

## Detection of harmonic components

To measure the performance of VocalMat for the detection of harmonic components, we compared the output of VocalMat with the *test* data set. The rate of true positives was 93.32 ± 1.96% (mean ± SEM; median = 92.18; 95% CI [88.54, 98.11]). The rate of USVs wrongly labeled as having a harmonic component (false positive) was 5.39 ± 1.18% (mean ± SEM; median = 5.17; 95% CI [2.50, 8.27]). The rate of missed harmonic components (false negative) was 6.68 ± 1.96% (mean ± SEM; median = 7.82, 95% CI [1.89, 11.46]). All combined, the error rate in identifying harmonic components was 12.19 ± 3.44% (mean ± SEM; median = 11.92, 95% CI [3.34, 21.03]). Thus, VocalMat presents satisfactory performance in detecting the harmonic components of the USVs.

## Classification of USVs in categories

To evaluate the performance of VocalMat in classifying the detected USVs in distinct categories, we compared the most likely label assigned by the CNN to the labels assigned by the investigators (i.e.,

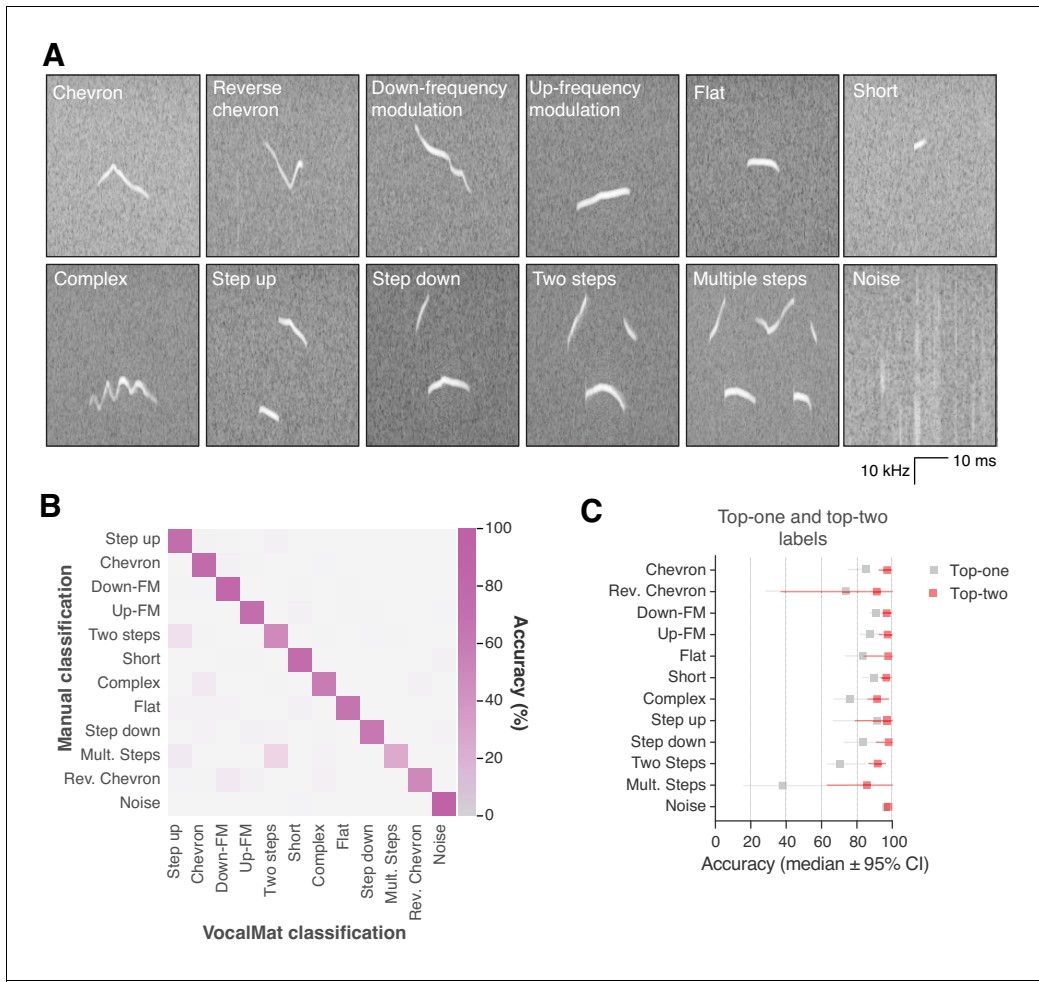

**Figure 4.** VocalMat performance for ultrasonic vocalization (USV) classification. (**A**) Example of the 11 categories of USVs plus noise that VocalMat used to classify the USV candidates. (**B**) Confusion matrix illustrating VocalMat's performance in multiclass classification (see also **Supplementary file 5** and **Figure 4—figure supplement 1** for individual confusion matrices). (**C**) Comparison of classification performance for labels assigned based on the most likely label (Top-one) versus the two most likely labels (Top-two) (see **Supplementary file 6**). Symbols represent median ±95% confidence intervals.

The online version of this article includes the following figure supplement(s) for figure 4:

**Figure supplement 1.** Confusion matrix illustrating VocalMat's performance in multiclass classification per recording file.

ground-truth). The accuracy of the VocalMat classifier module is 86% (*Figure 4B,C*, *Figure 4—figure supplement 1*, and *Supplementary file 5*). VocalMat shows lower accuracy to detect rare USV types (e.g., reverse chevron; *Figure 4A–C*) or USVs with multiple components (e.g., multiple steps and two steps; *Figure 4A–C*). When we expanded our analysis to consider the two most likely labels assigned by the CNN, the accuracy of VocalMat was 94% (*Figure 4E* and *Supplementary file 6*). These observations suggest a possible overlap between the definition of categories. Based on these analyses, we reasoned that the distribution of probabilities for each of the 11 categories of USV types calculated by the CNN could provide a more fluid classification method to analyze the vocal repertoire of mice.

## Using VocalMat to analyze and visualize the vocal repertoire of mice

To illustrate the use of the probability distribution of USV classification by VocalMat, we used data previously published by our group with over 45,000 USVs (*Zimmer et al., 2019*). In this published data set, two groups of 10 days old mice were studied. At this age, mice vocalize in the ultrasonic range when separated from the nest. Two groups of mice were analyzed (control versus treatment) during two contiguous time points (baseline versus test). The difference between the two groups was that in the treatment group, a specific population of neurons in the brain was activated to induce higher rates of USV emission (*Zimmer et al., 2019*).

To visualize the probability distribution of USV classification by VocalMat, we used Diffusion Maps (see Materials and methods). Diffusion Maps is a dimensionality reduction algorithm that allows the projection of the probability distribution into a Euclidean space (*Coifman et al., 2005*). We compared all four experimental conditions against each other and visually verified that the manifolds representing the USV repertoires showed a degree of similarity (*Figure 5A*).

To quantify the similarities (or differences) between the manifolds, we calculated the pairwise distance between the centroids of USV types within each manifold (*Figure 5B*). The pairwise distance matrices provide a metric for the manifold structure, allowing a direct comparison between the vocal repertoire of different groups. When we compared the similarity between the pairwise distance matrices in the four experimental conditions, we observed that the treatment group in the test condition presented a robust structural change in the vocal repertoire, which can be effectively represented by a matrix correlation (*Figure 5C*). The degree of similarity between the experimental conditions can also be visualized by comparing the structure of the manifolds. Since the manifolds are calculated separately, their coordinate system needs to be aligned to allow visual comparisons, which we achieve using the Kernel Alignment algorithm (*Figure 5—figure supplement 1* and Materials and methods) (*Tuia and Camps-Valls, 2016*; *Wang and Mahadevan, 2011*). The quality of the manifold alignment is assessed by Cohen's coefficient and overall projection accuracy into a joint space (*Figure 5—figure supplement 1*), showing the lowest scores for the treatment group in the test condition when compared to the other experimental conditions. Hence, these later analyses illustrate the use of the probability distribution for vocal classification and the power of dimensionality reduction techniques – such as Diffusion Maps – to provide a detailed analysis of the vocal repertoire of mice.

## Discussion

The premise of ethology is that to *understand* how the brain works, it is first important to quantify behavior (*Pereira et al., 2020*; *Tinbergen, 1963*). The vocal behavior of animals is especially tractable for precise quantification as sound waves can be recorded with extreme resolution in multiple dimensions (time, frequency, intensity). Methods for quantifying vocal behavior, however, are often dependent on labor-intense customization and parameter tuning. Here, we reported the development of VocalMat, a software to automatically detect and classify mouse USVs with high sensitivity. VocalMat eliminates noise from the pool of USV candidates, preserves the main statistical components for the detected USVs, and identifies harmonic components. Additionally, VocalMat uses machine learning to classify USV candidates into 11 different USV categories. VocalMat is open-source, and it is compatible with high-performance computing clusters that use the Slurm job scheduler, allowing parallel and high-throughput analysis.

VocalMat adds to the repertoire of tools developed to study mouse USVs (*Van Segbroeck et al., 2017*; *Burkett et al., 2015*; *Chabout et al., 2015*; *Arriaga et al., 2012*; *Holy and Guo, 2005*;

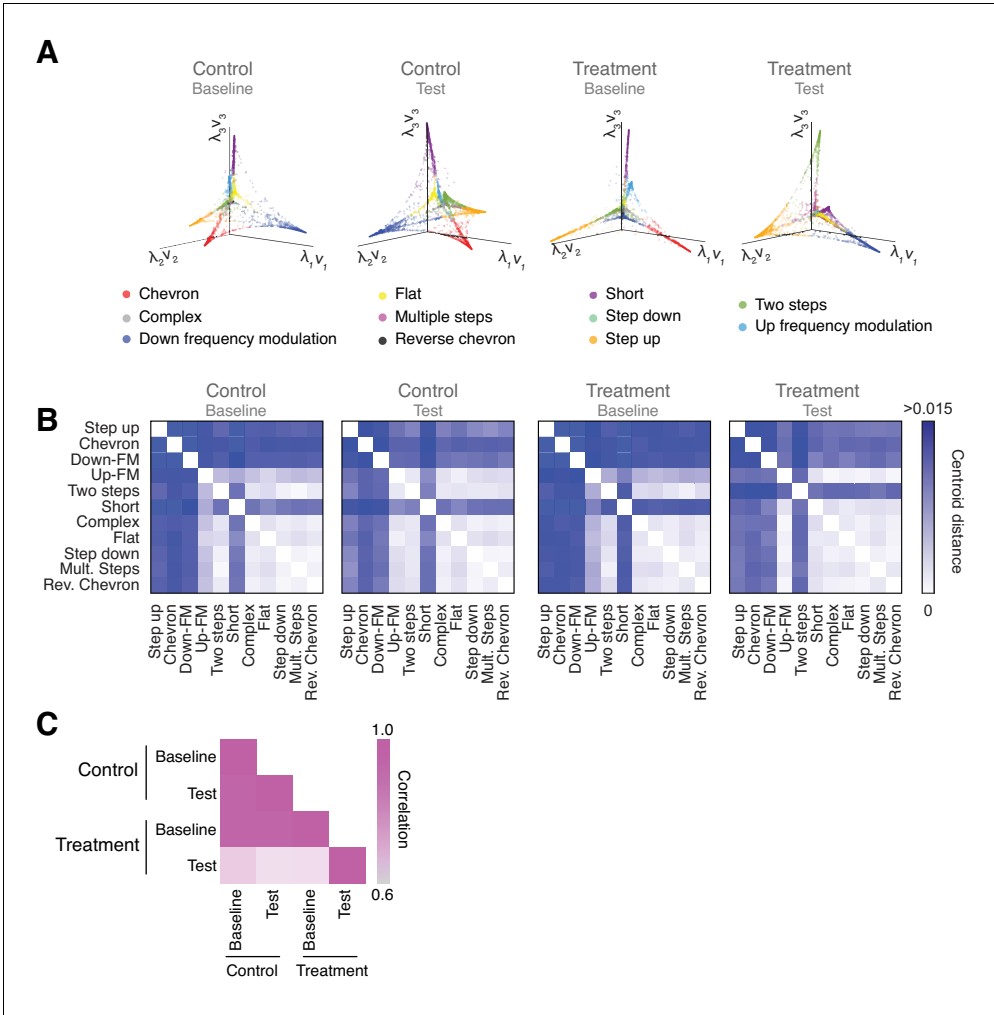

**Figure 5.** Vocal repertoire visualization using Diffusion Maps. (**A**) Illustration of the embedding of the ultrasonic vocalizations (USVs) for each experimental condition. The probability distribution of all the USVs in each experimental condition is embedded in a Euclidean space given by the eigenvectors computed through Diffusion Maps. Colors identify the different USV types. (**B**) Pairwise distance matrix between the centroids of USV types within each manifold obtained for the four experimental conditions. (**C**) Comparison between the pairwise distance matrices in the four experimental conditions by Pearson's correlation coefficient.

The online version of this article includes the following figure supplement(s) for figure 5:

**Figure supplement 1.** Alignment of the manifolds between pairs of experimental conditions.

---

*Coffey et al., 2019*; *Tachibana et al., 2020*). These other tools depend on several parameters defined by the user, so it is difficult to compare their performance to VocalMat effectively. Nevertheless, our tests show that VocalMat outperforms other tools in both sensitivity and accuracy in detecting USVs. More importantly, VocalMat uses differential geometry to automate the task of USV detection that in combination with CNNs maximizes sensitivity without sacrificing accuracy without the need for *any* customization or parameter tuning.

Moreover, VocalMat provides a flexible classification method by treating USV classification as a problem of probability distribution across different USV categories. This approach allows the analysis, visualization, and comparison of the repertoires of USVs of different mice and experimental groups using dimensionality reduction algorithms.

VocalMat uses a pattern recognition approach, based on CNNs, which learns directly from the training set without the need for feature extraction via segmentation processes (*Schmidhuber, 2015*; *Krizhevsky et al., 2012*). This characteristic provides the possibility for adaptability of VocalMat to

different experimental settings, including its use with other species and vocal types. Because Vocal-Mat preserves the features of the detected USVs – including temporal dynamics, frequency, intensity, and morphology – it also provides a rich data set to quantify vocal behavior.

In summary, VocalMat is a tool to detect and classify mouse USVs with outstanding sensitivity and accuracy while keeping all the relevant spectral features, including harmonic components. With VocalMat, USVs can be quantified with precision, thus creating the opportunity for a more detailed understanding of this behavior.

# Materials and methods

## Animals

All mice used to record the emission of USV were 5–15 days old from both sexes. Dams used were 2–6 months old and were bred in our laboratory. To maximize variability in the audio recordings, different mouse strains were used (all from The Jackson Laboratories): C57Bl6/J, NZO/HlLtJ, 129S1/SvImJ, NOD/ShiLtJ, and PWK/PhJ. Each mouse was only recorded once. All mice were kept in temperature- and humidity-controlled rooms, in a 12/12 hr light/dark cycle, with lights on from 7:00 AM to 7:00 PM. Food and water were provided ad libitum. All procedures were approved by the IACUC at Yale University School of Medicine.

## Audio acquisition

Mice were recorded in different conditions to maximize the levels of environmental noise and other experimental factors in the audio recordings. Mice were placed in a small box made of Plexiglas (15 × 15 × 10 cm). This small box was placed inside an open field (40 × 40 × 40 cm) with the walls covered by anechoic material (2' Wedge Acoustic Foam, Auralex) or inside an environmental chamber (CAB-8, Sable System International). For the audio recordings from inside the environmental chamber, the heater was either turned on at 35°C or turned off, as summarized in *Table 3*. By turning the heater on, we increased the levels of ambient noise to train and test VocalMat in a wide range of conditions. Sound was recorded using the recorder module UltraSoundGate 416H and a condenser ultrasound microphone CM16/CMPA (Avisoft Bioacoustics, Berlin, Germany) placed 15 cm above the animal. The experiments were recorded with a sampling rate of 250 kHz. The recording system had a flat response for sounds within frequencies between 20 kHz and 140 kHz, preventing distortions for the frequency of interest. The recordings were made by using Avisoft RECORDER 4.2 (version 4.2.16; Avisoft Bioacoustics) in a Laptop with an Intel i5 2.4 GHz processor and 4 GB of RAM. Using these settings, 10 min of audio recording generated files of approximately 200 MB.

## Spectral power

USVs were segmented on the audio files by analysis of their spectrograms. Aiming the configuration that would grant us the best time-frequency resolution, the spectrograms were calculated through a short-time Fourier transformation (STFT, *MATLAB*'s spectrogram function) using the following parameters: 1024 sampling points to calculate the discrete Fourier transform (NFFT = 1024), Hamming window with length 256, and half-overlapping with adjacent windows to reduce artifacts at the

**Table 3.** Summary of experimental conditions covered in the *test* data set.

| Age | Microphone gain | Location | Heating |
|---|---|---|---|
| P9 | Maximum | Environmental chamber | No |
| P9 | Maximum | Environmental chamber | No |
| P9 | Maximum | Environmental chamber | No |
| P10 | Intermediary | Open field | No |
| P10 | Intermediary | Open field | No |
| P10 | Maximum | Environmental chamber | Yes |
| P10 | Maximum | Environmental chamber | Yes |

boundary. The mathematical expression that gives us the short-time Fourier Transform is shown below:

$$STFT\{x[n]\}(m,\omega) = X(n,\omega) = \sum_{n=-\infty}^{\infty} = x[n]w[n-m]e^{-j\omega n} \tag{1}$$

where the original signal $x[n]$ is divided in chunks by the windowing function $w[m]$. The Fourier Transformation of the chunks result in a matrix with magnitude and phase for each time-frequency point.

The spectral power density, represented in the logarithmic unit decibels, is then given by

$$P(m,\omega) = 10\log\left|\sum_{m=-\infty}^{\infty} x[m]w[n-m]e^{-j\omega n}\right|^2 \tag{2}$$

We used a high pass filter (45 kHz) to eliminate sources of noise in the audible range and to reduce the amount of data stored (*Grimsley et al., 2011*).

## Normalization and contrast enhancement

Since USVs present higher intensity than the background and to avoid setting a fixed threshold for USV segmentation, we used contrast adjustment to highlight putative USV candidates and to reduce the variability across audio files. Contrast adjustment was obtained according to the following rescaling equation:

$$J = \left(\frac{\frac{|10log(P)|}{max(10log(P))} - L_{in}}{H_{in} - L_{in}}\right)^{\gamma} \tag{3}$$

where $H_{in}$ and $L_{in}$ are the highest and the lowest intensity values of the adjusted image, respectively, and $P$ is the power spectrum for each time-frequency point (pixel of the spectrogram). The parameter $\gamma$ describes the shape of the mapping function between the original and the corrected image, such that $\gamma<1$ results in darker pixels and $\gamma>1$ in brighter pixels. We used a linear mapping for our application ($\gamma = 1$, *MATLAB*'s imadjust function).

## Adaptive thresholding and morphological operations

Due to non-stationary background noise and dynamic changes in the intensity of USVs within and between the audio files, we use adaptive thresholding methods to binarize the spectrograms. The threshold is computed for each pixel using the local mean intensity around the neighborhood of the pixel (*Bradley and Roth, 2007*). This method preserves hard contrast lines and ignores soft gradient changes. The integral image consists of a matrix $I(x,y)$ that stores the sum of all pixel intensities $f(x,y)$ to the left and above the pixel $(x,y)$. The computation is given by the following equation:

$$I(x,y) = f(x,y) + I(x-1,y) + I(x,y-1) - I(x-1,y-1) \tag{4}$$

Therefore, the sum of the pixel values for any rectangle defined by a lower right corner $(x_2,y_2)$ and upper left corner $(x_1,y_1)$ is given as:

$$\sum_{x=x_1}^{x_2}\sum_{y=y_1}^{y_2} f(x,y) = I(x_2,y_2) - I(x_2,y_1-1) + I(x_1-1,y_2) - I(x_1-1,y_1-1) \tag{5}$$

Then, the method computes the average of an $s \times s$ window of pixels centered around each pixel. The average is calculated considering neighboring pixels on all sides for each pixel. If the value of the current pixel intensity is $t$ percent less than this average, then it is set to black; otherwise it is set to white, as shown in the following equation:

$$C(x,y) = \frac{1}{(y_2-y_1)(x_2-x_1)} \cdot \sum_{x=x_1}^{x_2}\sum_{y=y_1}^{y_2} f(x,y) \tag{6}$$

where $C(x,y)$ represents the average around the pixel $(x,y)$.

The binarized image is then constructed such as that pixels $(x, y)$ with intensity $t$ percent lower than $C(x, y)$ are set to black (*Bradley and Roth, 2007*):

$$B(x,y) = \begin{cases} 0, & \text{if } f(x,y) \leq (1-t)C(x,y) \\ 1, & \text{otherwise} \end{cases} \tag{7}$$

where $t$ represents the sensitivity factor, and it was empirically chosen as $t = 0.2$ for our application. The resulting binarized image consists of background pixels (intensity = 0) and putative vocal segments (contiguous areas with intensity = 1). The segments are then subjected to a sequence of morphological operations: (i) opening (erosion followed by a dilation; *MATLAB*'s imopen) with a rectangle 4 × 2 pixels as kernel; (ii) dilation with a line of length $l = 4$ and $\angle$ relative to the horizontal axis as kernel (*MATLAB*'s imdilate); (iii) filtering out candidates (i.e., dense set of white pixels) with <60 pixels (correspondent to approximately 2 ms syllable); and (iv) dilation with a line of length $l = 4$ and $\angle\, 0°$, making the USV candidates proportional to their original shape.

## Local median filter

The image processing pipeline used for segmentation can create artifacts or noise that were not originally present in the spectrogram, especially in the binarization step. These noises that occur due to the segmentation process are not associated with an event in the recording (a real USV or external noise) and are part of the pool of USV candidates. To determine if a USV candidate is relevant for further analysis, we used a contrast filter – Local Median Filter – to compare the median intensity of the pixels in the USV candidate $k$ (referred to as $\widehat{X}_k$) to the intensity of the pixels in a bounding box that encompasses the USV candidate (referred to as $\widehat{W}_k$). The Local Median Filter then determines if a USV candidate $k$ is discarded based on the cumulative distribution of intensity ratio over all the USV candidates detected in the audio file $\widehat{X}/\widehat{W}$. The bounding box that defines the window $W_k$ is a rectangle with its four vertices defined as a function of the frequencies ($F_k$) for USV candidate $k$ and its time stamps ($T_k$). Thus, the bounding box is defined as follows:

$$W_k = \begin{cases} (max(F_k) + 2.5)kHz, \\ (min(F_k) - 2.5)kHz, \\ (max(T_k) + 0.1)s, \\ (min(T_k) - 0.1)s \end{cases} \tag{8}$$

As seen in (8), a 200 ms interval is analyzed around the USV candidate. Such a wide interval may present more than one USV in $W_k$. However, the amount of pixels in $X_k$ represents only 2.43 ± 0.10% (mean ± SEM; median = 1.27, 95% CI [2.22 , 2.63]; n = 59,781 images analyzed) of the total number of pixels contained in the window $W_k$. Given this proportion between the number of pixels in $X_k$ and $W_k$, the median of the intensity distribution of the whole window ($\widehat{W}_k$) tends to converge to the median intensity of the background.

We used the ratio $C_k = \widehat{X}_k/\widehat{W}_k$ to exclude USV candidates that correspond to segmentation noise. We first calculated the CDF of $C_k$ over all the USV candidates in an audio file (now referred to as $\Upsilon$). To find the inflection point in $\Upsilon$, a second-order polynomial fit for every set of three consecutive points was used to obtain local parametric equations ($\Upsilon(t) = (x(t), y(t))$) describing the segments of $\Upsilon$. Since the calculation of the inflection point is done numerically, the number of points chosen for this calculation should be such that we can have as many points of curvature as possible while preserving information of local curvature. Then, after a screening for the best number of points, $\Upsilon$ was down-sampled to 35 equally spaced points and the inflection point was calculated. Using the local parametric equations, we calculated the tangent and normal vectors on each of the 35 points. Using these vectors, we estimated the changing rate of the tangent toward the normal at each point, which is the curvature $\kappa$ (*O'neill, 2006*) and can be calculated as follows:

$$\kappa = \frac{\det(\Upsilon', \Upsilon'')}{\Upsilon'^3} \tag{9}$$

or by using the parametric equations:

$$\kappa = \frac{x'y'' - x''y'}{(x^2 + y^2)^{3/2}} \tag{10}$$

The inflection point is then determined as the point with maximum curvature of the CDF curve, and adopted as threshold τ. This threshold is calculated individually for each audio file since it can vary according to the microphone gain and the distance of the microphone from the sound source. In audio files with a very low number of USVs, the point of maximum curvature of the CDF curve was not detected, and no τ was estimated. In these cases, a default threshold $\tau = 0.92$ was adopted as a conservative threshold, since no audio file presented inflection point as high as 0.92 in our *training* set. Only the USV candidates satisfying (*Equation 11*) are kept for further analysis.

$$\left\{ X_k \in \chi | \widehat{X_k} \leq \tau \widehat{W_k} \right\}$$

$$\tag{11}$$

where χ represents the set of USV candidates that survived the Local Median Filter. Of note, the intensity of each pixel is calculated in decibels, which is given in negative units due to the low power spectrum.

## CNNs for USV classification

We use CNNs to eliminate external noise from the pool of USV candidates and classify USVs in distinct types (see below). We use a transfer learning approach with an AlexNet (*Krizhevsky et al., 2012*) model pre-trained on the ImageNet data set, and perform end-to-end training using our USVs data set. Briefly, the last three layers of the network were replaced in order to handle a 12-categories classification task for our data set (11 *USV types + noise*).

The outputs of the segmentation process with detected USV candidates were centralized in windows of 220 ms. These windows were twice the maximum duration of USVs observed in mice (*Grimsley et al., 2011*) and were framed in individual 227 × 227 pixels images. Each image was then manually labeled by an experienced experimenter as noise (including acoustic or segmentation noise) or one of the USV categories. The labeled data set was used to train the CNN to classify the USV candidates.

The images in our data set were manually labeled based on definitions for USV classes found in previous studies (adapted from *Scattoni et al., 2008* and *Grimsley et al., 2011*). The USV classes are described below:

### Complex
One-note syllables with two or more directional changes in frequency >6 kHz. A total of 350 images were used for training.

### Step up
Two-notes syllables in which the second element was ≥6 kHz higher from the preceding element and there was no more than 10 ms between steps. A total of 1814 images were used for training.

### Step down
Two-notes syllables in which the second element was ≥6 kHz lower from the preceding element and there was no more than 10 ms between steps. A total of 389 images were used for training.

### Two steps
Three-notes syllables, in which the second element was ≥6 kHz or more different from the first, the third element was ≥6 kHz or more different from the second and there was no more than 10 ms between elements. A total of 701 images were used for training.

### Multiple steps

Four-notes syllables or more, in which each element was ≥6 kHz or more different from the previous one and there was no more than 10 ms between elements. A total of 74 images were used for training.

### Up-frequency modulation

Upwardly frequency modulated with a frequency change ≥6 kHz. A total of 1191 images were used for training.

### Down-frequency modulation

Downwardly frequency modulated with a frequency change ≥6 kHz. A total of 1775 images were used for training.

### Flat

Constant frequency syllables with modulation ≤5 kHz and duration ≥12 ms. A total of 1134 images were used for training.

### Short

Constant frequency syllables with modulation ≤5 kHz and duration ≤12 ms. A total of 1713 images were used for training.

### Chevron

Shaped like an inverted *U* in which the peak frequency was ≥6 kHz than the starting and ending frequencies. A total of 1594 images were used for training.

### Reverse chevron

Shaped like an *U* in which the peak frequency was ≥6 kHz than the starting and ending frequencies. A total of 136 images were used for training.

### Noise

Any sort of mechanical or segmentation noise detected during the segmentation process as a USV candidate. A total of 2083 images were used for training.

In order to purposely create some overlap between the categories, USVs with segments oscillating between 5 and 6 kHz were not defined or used for training. The assumption is that the CNN should find its transition method between two overlapping categories.

Our *training* data set consisted of 12,954 images, wherein 2083 were labeled as noise. This data set correspond to mice of different strains (C57Bl6/J, NZO/HlLtJ, 129S1/SvImJ, NOD/ShiLtJ, and PWK/PhJ) and ages (postnatal day (P)5, P10, and P15) from both sexes.

The CNN was trained using stochastic gradient descent with momentum, a batch size of M = 128 images, and with a maximum number of epochs set to 100. Through a screening process for the set of hyper-parameters that would maximize the average performance of the network, the chosen learning rate was $\alpha = 10^{-4}$, momentum of 0.9, and weight decay $\lambda = 10^{-4}$. To validate the training performance, each data set was randomly split into two disjoint sets; *training* set (90%) and a *validation* set (10%). The training and validation sets were independently shuffled at every epoch during training. The training was set to stop when the classification accuracy on the validation set did not improve for three consecutive epochs. When running in a GeForce GTX 980 TI, the final validation accuracy was 95.28% after 17 min of training.

## Testing detection performance

To evaluate the performance of VocalMat, neonatal mice were recorded for 10 min upon social isolation in different conditions (*Table 3*) to increase the variability of the data. To cover most of the recording setups, we included recordings with different microphone settings (maximum and intermediary gain), environments (open field or enclosed environmental chamber), and machinery noise (heater on or off). The spectrograms were manually inspected for the occurrence of USVs. The

starting time for the detected USVs was recorded. USVs automatically detected by VocalMat with a start time matching manual annotation (±5 ms of tolerance) were considered correctly detected. USVs manually detected with no correspondent USV given by VocalMat were considered *false negative*. The false negatives were originated from missed USVs or USVs that the software labeled as noise. Finally, USVs registered by VocalMat without a correspondent in the manual annotation were considered *false positive*. In order to compare VocalMat to the other tools available, the same metrics were applied to the output of Ax (*Neunuebel et al., 2015*), MUPET (*Van Segbroeck et al., 2017*), USVSEG (*Tachibana et al., 2020*), and DeepSqueak (*Coffey et al., 2019*).

## Diffusion maps for USV class distribution visualization

One of the main characteristics of VocalMat is attributing to USVs a distribution of probabilities over all the possible vocal classes. Since we classify USV candidates in 11 categories, to have access to the distribution of probabilities we would need to visualize the data in 11 dimensions. Here, as an example of analytical methods that can be applied to the output data from VocalMat, we used Diffusion Maps (*Coifman et al., 2005*) to reduce the dimensionality of the data to three dimensions. Diffusion Maps allow remapping of the data into a Euclidean space, which ultimately results in preserving the distance between USVs based on the similarity of their probability distribution. A Gaussian kernel function defines the connectivity between two data points in a Euclidean manifold. Such kernel provides the similarity value between two data points $i$ and $j$ as follows:

$$W_{ij} = \exp\left(\frac{-x_i - x_j^2}{2\sigma^2}\right) \tag{12}$$

where $W_{ij}$ represents the similarity value between observations $i$ and $j$. The parameter $\sigma$ corresponds to the width of the kernel, and it is set based on the average Euclidean distance observed between observations of the same label (i.e., the intra-class distances). For our application, $\sigma = 0.5$ was set based on the distance distribution observed in our data.

The similarity matrix is then turned into a probability matrix by normalizing the rows:

$$p(j|i) = \frac{W_{ij}}{\sum_k W_{ik}} = D^{-1}W = M_{ij} \tag{13}$$

where $\sum_k W_{ik} = D_{ii}$ has the row sum of $W$ along its diagonal. The matrix $M$ gives the probability of going from node $i$ to any other node using a random walk. In other words, the probability that the USV $i$ is close to another USV $j$ given their probability distribution.

Once we take one step in such Euclidean space, the probabilities are updated, since the set of likely nodes for the next move are now different. This idea of moving from node to node while updating the probabilities results in a 'diffused map'.

The process of moving from a USV $i$ to $j$ after $t$ steps in this Euclidean space is computed as follows:

$$p(t,j|i) = e_i^T M^t e_j \tag{14}$$

For our application, we use $t = 2$.

Next, we find the coordinate functions to embed the data in a lower-dimensional space. The eigenvectors of $M$ give such a result. Because $M$ is not symmetric, the eigendecomposition is computed through a SVD decomposition (*Golub and Kahan, 1965*):

$$M_s := D^{1/2}MD^{-1/2} = D^{1/2}D^{-1}WD^{-1/2} = D^{-1/2}WD^{-1/2} \tag{15}$$

and since $D^{-1/2}$ and $W$ are symmetric, $M_s$ is also symmetric and allows us to calculate its eigenvectors and eigenvalues. For the sake of notation, consider:

$$M_s = \Omega\Lambda\Omega^T \Longrightarrow M = D^{-1/2}\Omega\Lambda\Omega^T D^{1/2} \tag{16}$$

Considering $\Psi = D^{-1/2}\Omega$ (right eigenvectors of $M$) and $\Phi = D^{1/2}\Omega$ (left eigenvectors of $M$), we verify that $\Phi^T = \Psi^{-1}$, therefore they are mutually orthogonal and $M$ and $M_s$ are similar matrices. Thus,

$$M = \Psi\Lambda\Psi^{-1} = \Psi\Lambda\Psi^T \tag{17}$$

and the diffusion component shown in **Equation (14)** is incorporated as the power of the diagonal matrix composed by the eigenvalues of $M$:

$$M^t = \Psi \Lambda^t \Phi^T \tag{18}$$

We use the scaled right eigenvectors by their corresponding eigenvalues ($\Gamma = \Psi \Lambda$) as the coordinate functions. Since the first column of $\Gamma$ is constant across all the observations, we use the second to fourth coordinates in our work.

## Vocal repertoire analysis via manifold alignment

The result of the embedding by Diffusion Maps allows 3D visualization of the probability distribution for the USVs. The direct comparison of different 3D maps is challenging to obtain as the manifolds depend on data distribution, which contains high variability in experimental samples. To address this problem and compare the topology of different manifolds, we used a manifold alignment method for heterogeneous domain adaptation (**Wang and Mahadevan, 2011**; **Tuia and Camps-Valls, 2016**). Using this method, two different domains are mapped to a new latent space, where samples with the same label are matched while preserving the topology of each domain.

We used the probability distribution for the USVs for each data set to build the manifolds (**Wang and Mahadevan, 2011**). Each manifold was represented as a Laplacian matrix constructed from a graph that defines the connectivity between the samples in the manifold. The Laplacian matrix is then defined as $L = W_{ij} - D_{ii}$ (see **Equation (12)**).

The final goal is to remap all the domains to a new shared space such that samples with similar labels become closer in this new space. In contrast, samples with different labels are pushed away while preserving the geometry of the manifolds. It leads to the necessity of three different graph Laplacians: $L_s$ (relative to the similarity matrix and responsible for connecting the samples with the same label), $L_d$ (dissimilarity matrix and responsible for connecting the samples with different labels), and $L$ (similarity matrix responsible for preserving the topology of each domain). **Wang and Mahadevan, 2011** show that the embedding that minimizes the joint function defined by the similarity and dissimilarity matrices is given by the eigenvectors corresponding to the smallest nonzero eigenvalues of the following eigendecomposition:

$$Z(L + \mu L_s)Z^T V = \lambda Z L_d Z^T V \tag{19}$$

where $Z$ is a block diagonal containing the data matrices $X_i \in \mathbb{R}^{d_i \times n_i}$, (where $n_i$ samples and $d_i$ dimensions are constants for the $i^{th}$ domain) from the two domains. Thus, $Z = diag(X_1, X_2)$. The matrix $V$ contains the eigenvectors organized in rows for each domain, $V = [v_1, v_2]^T$. The $\mu$ is weight parameter, which goes from preserving both topology and instance matching equally ($\mu = 1$) or focusing more on topology preservation ($\mu > 1$) (**Tuia and Camps-Valls, 2016**).

From **Equation (19)**, we then extract $N_f = \sum_{i=1}^D d_i$ features, which is the sum of the dimensions of the individual domains (see details in **Wang and Mahadevan, 2011**; **Tuia and Camps-Valls, 2016**), and the projection of the data to a joint space $\mathcal{F}$ defined by the mapping matrix $V$ will be given by

$$P_{\mathcal{F}}(X_i) = v_i^T X_i \tag{20}$$

To measure the performance of the alignment, linear discriminant analysis (LDA) (**McLachlan, 2004**) is used to show the ability to project the domains in a joint space. The LDA is trained on half of the samples in order to predict the other half. The error of the alignment is given as the percentage of samples that would be misclassified when projected into the new space (overall accuracy) (**Tuia and Camps-Valls, 2016**).

Another measurement to quantify the quality of the alignment is by calculating the agreement between the projections, which is given by Cohen's Kappa coefficient ($\kappa$) (**Agresti, 2018**). In this method, the labels are treated as categorical, and the coefficient compares the agreement with that expected if ratings were independent. Thus, disagreements for labels that are close are treated the same as labels that are far apart.

Cohen's coefficient is defined as:

$$\kappa = \frac{p_0 - p_e}{1 - p_e} \tag{21}$$

where $p_0$ is the observed agreement ($p_0 = \sum_{i=1}^{k} p_{ii}$ for a confusion matrix $p = n/N$, in which $n$ is the raw confusion matrix and $N$ is the total number of samples, composed by the projection of the $k$ labels), which corresponds to the accuracy; $p_e$ is the probability of agreement by chance ($p_e = \frac{1}{N^2} \sum_{i=1}^{k} p_{i.} p_{.i}$, where $p_{i.}$ is the number of times an entity of label $i$ was labeled as any category and $p_{.i}$ is the number of times any category was predicted as label $i$). Therefore, a $\kappa = 0$ represents no agreement (or total misalignment of manifolds) and $\kappa = 1$ is a total agreement.

In this context, the overall accuracy ($OA$) is given by $OA = \sum_{i=1}^{k} p_{ii}/N$, where $N$ is the total number of samples.

The asymptotic variance for $\kappa$ is given as follows:

$$\hat{\sigma}^2(\hat{\kappa}) = \frac{1}{N}\left[\frac{\theta_1(1-\theta_1)}{(1-\theta_2)^2} + \frac{2\theta_1(1-\theta_1)(2\theta_1\theta_2 - \theta_3)}{(1-\theta_2)^3} + \frac{(1-\theta_1)^2(\theta_4 - 4\theta_2^2)}{(1-\theta_2)^4}\right] \tag{22}$$

where

$$\theta_1 = \frac{1}{n}\sum_{i=1}^{k} n_{ii} \tag{23}$$

(which turns into accuracy once it is divided by $N$),

$$\theta_2 = \frac{1}{n^2}\sum_{i=1}^{k} n_{i.} n_{.i} \tag{24}$$

$$\theta_3 = \frac{1}{n^2}\sum_{i=1}^{k} n_{ii}(n_{i.} + n_{.i}) \tag{25}$$

$$\theta_4 = \frac{1}{n^3}\sum_{i=1}^{k}\sum_{j=1}^{k} n_{ij}(n_{j.} + n_{.i})^2 \tag{26}$$

From *Equation (22)* we can calculate the Z-score, which can express the significance of our $\kappa$:

$$Z = \frac{\kappa}{\hat{\sigma}^2(\hat{\kappa})} \tag{27}$$

And the 95% confidence interval as

$$CI = [\kappa + 1.96\sqrt{\hat{\sigma}^2(\hat{\kappa})}, \kappa - 1.96\sqrt{\hat{\sigma}^2(\hat{\kappa})}] \tag{28}$$

The third form of error measurement is the evaluation of the projection per USV class from each domain remapped into the new space. This method is based on the fact that this new space is the one in which the cost function expressed by *Equation (19)* is minimized and, therefore, the projection from each domain into the new space has its projection error for each class. As a consequence, the mean of the projection error from each domain to the new space for each class can be used as a quantitative measurement of misalignment of projected domains.

## Quantification and statistical analysis

MATLAB (2019a or above) and Prism 8.0 were used to analyze data and plot figures. All figures were created using Adobe Illustrator CS6/CC. Data were first subjected to a normality test using the D'Agostino and Pearson normality test or the Shapiro–Wilk normality test. When homogeneity was assumed, a parametric analysis of variance test was used. The Student's t-test was used to compare two groups. The Mann–Whitney U-test was used to determine the significance between groups. Two sample Kolmogorov–Smirnov test was used to calculate the statistical differences between the contrast of USVs and noise. Statistical data are provided in text and in the figures. In the text, values

are provided as mean ± SEM. p<0.05 was considered statistically significant. The 95% confidence intervals are reported in reference to the mean. The true positive rate is computed as the ratio between true positive (hit) and real positive cases. The true negative rate is the ratio between true negative (correct rejection) and real negative cases. The false negative rate is the ratio between false negative (type I error) and real positives cases. The false positive (type II error) is the ratio between false positive and real negative cases. The false discovery rate is the ratio between false positive and the sum of false positives and real positives.

### Code and data availability

VocalMat is available on GitHub (https://github.com/ahof1704/VocalMat.git; *Fonseca, 2021*; copy archived at swh:1:rev:9384fabfc1fbd9bc0ef8ca460b652e72c5b6819f) for academic use. Our data set of vocalization images is available on OSF (https://osf.io/bk2uj/).

## Acknowledgements

We thank the lab members for critical data collection and insights in the manuscript. MOD was supported by a NARSAD Young Investigator Grant ID 22709 from the Brain and Behavior Research Foundation, by the National Institute Of Diabetes And Digestive And Kidney Diseases of the National Institutes of Health (R01DK107916), by a pilot grant from the Yale Diabetes Research Center (P30 DK045735), by the Yale Center for Clinical Investigation Scholar Award, by the Whitehall Foundation, by the Charles H Hood Foundation, Inc (Boston, MA), by a pilot grant from the Modern Diet and Physiology Research Center (The John B Pierce Laboratory), by a grant of the Foundation for Prader-Willi Research, and by the Reginald and Michiko Spector Award in Neuroscience. MOD also received support from the Conselho Nacional de Desenvolvimento Científico e Tecnologico (CNPq) and Coordenadoria de Aperfeiçoamento de Pessoal de Nível Superior (CAPES), Brazil. AHOF and GMS were supported in part by the Coordenadoria de Aperfeiçoamento de Pessoal de Nível Superior – Brasil (CAPES) – Finance Code 001. GBO was supported in part by the Gilliam Fellowship for Advanced Studies from the HHMI. The authors declare no conflict of interest.

## Additional information

### Funding

| Funder | Grant reference number | Author |
| --- | --- | --- |
| National Institute of Diabetes and Digestive and Kidney Diseases | R01DK107916 | Marcelo O Dietrich |
| Brain and Behavior Research Foundation | NARSAD Young Investigator Grant ID 22709 | Marcelo O Dietrich |
| Whitehall Foundation | | Marcelo O Dietrich |
| Charles H. Hood Foundation | | Marcelo O Dietrich |
| Foundation for Prader-Willi Research | | Marcelo O Dietrich |
| Reginald and Michiko Spector Award in Neuroscience | | Marcelo O Dietrich |
| Yale Center for Clinical Investigation | | Marcelo O Dietrich |
| Yale Diabetes Research Center | P30DK045735 | Marcelo O Dietrich |
| Modern Diet and Physiology Research Center (The John B. Pierce Laboratory) | | Marcelo O Dietrich |
| Coordenação de Aperfeiçoamento de Pessoal de Nível Superior | Finance Code 001 | Antonio HO Fonseca Gustavo M Santana Sérgio Bampi Marcelo O Dietrich |

| Conselho Nacional de Desen-volvimento Científico e Tecno-lógico | | Sérgio Bampi<br>Marcelo O Dietrich |
| Howard Hughes Medical Institute | Gilliam Fellowship | Gabriela M Bosque Ortiz<br>Marcelo O Dietrich |

The funders had no role in study design, data collection and interpretation, or the decision to submit the work for publication.

## Author contributions

Antonio HO Fonseca, Conceptualization, Data curation, Formal analysis, Supervision, Validation, Investigation, Visualization, Methodology, Writing - original draft, Writing - review and editing; Gustavo M Santana, Conceptualization, Data curation, Formal analysis, Validation, Investigation, Visualization, Methodology, Writing - review and editing; Gabriela M Bosque Ortiz, Data curation, Validation; Sérgio Bampi, Data curation, Supervision, Funding acquisition, Methodology, Writing - review and editing; Marcelo O Dietrich, Conceptualization, Data curation, Formal analysis, Supervision, Funding acquisition, Validation, Investigation, Visualization, Methodology, Writing - original draft, Project administration, Writing - review and editing

## Author ORCIDs

Antonio HO Fonseca (iD) https://orcid.org/0000-0001-7791-8010
Gustavo M Santana (iD) https://orcid.org/0000-0003-1897-1625
Marcelo O Dietrich (iD) https://orcid.org/0000-0001-9781-2221

## Ethics

Animal experimentation: This study was performed in strict accordance with the recommendations in the Guide for the Care and Use of Laboratory Animals of the National Institutes of Health. The protocol was reviewed and approved by the Yale University Institutional Animal Care and Use Committee (IACUC). All of the animals were handled according to the approved IACUC protocol (#2018-20042) of the Yale University School of Medicine.

## Decision letter and Author response

Decision letter https://doi.org/10.7554/eLife.59161.sa1
Author response https://doi.org/10.7554/eLife.59161.sa2

# Additional files

## Supplementary files

- Supplementary file 1. List of parameters and performance of Ax.
- Supplementary file 2. List of parameters and performance for MUPET.
- Supplementary file 3. List of parameters and performance for USVSEG.
- Supplementary file 4. List of parameters and performance for DeepSqueak.
- Supplementary file 5. VocalMat accuracy per class.
- Supplementary file 6. VocalMat accuracy considering the two most likely labels.
- Transparent reporting form

## Data availability

All the data and code used in this work are publicly available and can be found at: https://osf.io/bk2uj/ and https://www.dietrich-lab.org/vocalmat.

The following dataset was generated:

| Author(s) | Year | Dataset title | Dataset URL | Database and Identifier |
|---|---|---|---|---|
| Fonseca AHO, Santana GM, Bosque Ortiz GM, Bampi Sr, Dietrich MO | 2020 | USV training set | https://osf.io/bk2uj/ | Open Science Framework, bk2uj |

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
