## [Decision Letter]

**Acceptance summary:**

The authors developed a new software ("VocalMat") to automatically detect and classify mouse ultrasonic vocalizations into distinct categories. The software is based on tools of image processing and neural network classification of spectrograms, that is useful to analyze large dataset of pup and adult USVs in various mouse models and experimental designs. All the datasets collected and the software source codes are freely accessible.

**Decision letter after peer review:**

Thank you for submitting your article "Analysis of ultrasonic vocalizations from mice using computer vision and machine learning" for consideration by *eLife*. Your article has been reviewed by 3 peer reviewers, and the evaluation has been overseen by a Reviewing Editor and Andrew King as the Senior Editor. The reviewers have opted to remain anonymous.

The reviewers have discussed the reviews with one another and the Reviewing Editor has drafted this decision to help you prepare a revised submission.

Summary:

This manuscript presents new tool to detect and classify mice ultrasonic vocalizations (USVs). The tool ( VocalMat) applies neural network technology for categorization of the various USVs to predetermined categories of pup calls. The paper in the form submitted seems to fit more as a methodology paper. Indeed, the authors state that the goal of their work is to: "create a tool with high accuracy for USV detection that allows for the flexible use of any classification method."

The paper is well written and presents a useful tool to identify and classify USVs of mice. However, the reviewers think that the authors did not provide enough supporting evidence to claim that their method is significantly superior to other tools in the literature that attempted USV classification. For example Vogel et al. (2019) [https://doi.org/10.1038/s41598-019-44221-3], reported very similar (85%) accuracy using more mainstream ML approaches than attempted in this study with CNNs.

Moreover, some of the reviewers were not convinced that the comparison to other tools was conducted in an unbiased and completely fair manner and that the approach described in this paper really represents a significant advantage over other tools. For example, two reviewers claim that the authors used DeepSqueak on their dataset without properly training it for this type of data, while their tool is specifically trained for it. Also, the reviewers expect to see a confusion matrix to assess model performance and establish whether the model does indeed replicate accurately classes (or how skewed it is with dominating classes).

Overall, all the reviewers agree that they would like to see a more rigorous attempt to validate the findings presented (ideally also on an external database) and proper (unbiased) comparison with other similar software, to justify the claim that VocalMat performance in classification of USVs is indeed superior and novel to the methods already in use.

If the authors wish to have the manuscript considered as a research paper and not in the form of methods paper they should change the focus of the paper and provide more data showing a novel biological application of their pup calls classification findings. If not, we will be happy to consider a suitably revised version of the manuscript for the Tools and Resources section of *eLife*.

For your convenience the detailed comments of the 3 reviewers are included below.

Reviewer #1:

In the manuscript entitled "Analysis of ultrasonic vocalizations from mice using computer vision and machine learning", Fonseca at al. present a novel computational tool for analysis of mice ultrasonic vocalizations (USVs). This tool aims to (1) detect USV candidates from audio clips; (2) separate the USVs from the noise; (3) apply neural network technology for categorization of the various USVs to predetermined categories; They use this tool to analyze a large dataset of pup calls and validate their tool as well as compare it with other computational tools published in the last decade. Finally, they show how they can use diffusion maps and manifold alignment to distinguish between calls of distinct groups of pups.

This tool is nice, but rather limited in its abilities and do not represent a conceptual or technical breakthrough. As for limitations, the tool presented here is designed, trained and validated for a specific type of murine calls (pup calls) and a predefined set of 11 categories, which may not cover all possible categories. As for technical advancement, the software combines criteria-based detection with neural network classification, similarly to previously published tools mentioned by the authors. Moreover, although the authors claim superiority of their software over published tools, I wasn't convinced that this comparison was conducted in an unbiased and completely fair manner and that their tool really represents a significant advantage on other tools. For example, they used DeepSqueak on their dataset without properly training it for this type of data, while their tool is specifically trained for it. Moreover, the success rates the authors report for other tools here are significantly lower than those reported by the relevant publications, and do not fit comparisons made by others. Also, I wasn't convinced that the software presented here will be working just as well as the authors claim on a distinct set of data coming from another lab and recorded in distinct conditions. My attempt to activate the software failed due to an error.

Reviewer #2:

Nice study overall, well-articulated and easy to follow. I also highly commend the authors for making all data + source code available.

1. The goal of the study could be encompassing a wider aim. The authors state: "Our goal was to create a tool with high accuracy for USV detection that allows for the flexible use of any classification method." The task of USV detection is relatively simple (no surprise the authors get 98% accuracy), it is the accurate classification of the USV types that is of particular importance. I would suggest the authors rephrase to emphasize that aspect.

2. "The output of the vocal classification provides the additional benefit of a probability distribution of vocal classes, allowing for the use of nonlinear dimensionality reduction techniques to analyze the vocal repertoire". Probably this needs some rephrasing into something like 'the output of the classifier enables the exploration of the probabilistic class membership'. More importantly though, that is not a pre-requisite for any dimensionality reduction techniques (linear or not). Dimensionality reduction could be directly applied to the extracted features, it is not dependent upon the classifier outputs.

3. "A linear regression analysis between manually validated data from different audio files and the true positives of the CNN revealed an almost-perfect linearity.…" I would expect to see a simple confusion matrix here assessing whether each of the USVs was correctly detected, rather than quantifying the number of USVs within each phonation (in which case indeed the methodology attempted by the authors would be appropriate). I think it is far more useful to assess on a case by case basis the USVs, and potentially determine whether e.g. one or more of the raw files was challenging for any reason. The authors could provide multiple confusion matrices 11x11 e.g. as a Supplementary Excel file.

4. "In order to evaluate the performance of VocalMat in detecting USVs compared to other published tools, we analyzed the same test dataset…" The authors' tool has an unfair advantage in this case, in that their algorithm was trained on the data collected under identical conditions like the test data. Moreover, the test data contains USVs from the same mouse.

5. "In summary, VocalMat is a new tool to detect and classify mouse USVs with superior sensitivity and accuracy while keeping all the relevant spectral features…" It is not clear if the authors mean superior to other studies in the literature that attempted USV classification. For example Vogel et al. (2019) [https://doi.org/10.1038/s41598-019-44221-3], reported very similar (85%) accuracy using more mainstream ML approaches than attempted in this study with CNNs.

6. If I understand correctly the methodology the authors used uses a single split of the data into training and testing, and data from a mouse could end up in both; thus the authors do not necessarily prove that their methods generalize well in new datasets. I would welcome the use of e.g. leave-one-mouse out assessment, and also the use of an additional external dataset collected under perhaps slightly different lab conditions (different lab) to see how well findings generalize. CNNs are extremely sensitive, and theoretical work has shown that adding imperceptible (visually) noise in images results in complete different results.

Reviewer #3:

The technical details should be better explained, with formulas and / or algorithm descriptions in one piece (not separated in main part and methods).

– Could you do a sensitivity analysis on their model for different number of observations?

– What is the efficiency of the method?

– Some state-of the art comparisons are missing: Did you compare it to the Automatic Mouse Ultrasound Detector? To DeepSqueak for classification? (This was only used for detection right?)

– What are the number of mice and number of records per mouse? How did you get them to vocalize?

– You have only used recordings of pups (5-15 d old). Do the results apply to adult mice?

– You used recordings of 5 strains of lab mice. Did you test or control for strain differences?

– There are many classifications that have been proposed, and so what was the basis for your syllable type classification? Can you please explain / motivate that more? Can mice discriminate these syllable types?

– How do you deal with the general machine learning problem that no insights are provided into the features that the model uses to classify syllable types ?

Is this method useful for recordings with a noisy background? Is there any reason to suspect that it might not work?

page1

The comments on the Abstract

"…detected more than >98% of the USVs" – What is the ground truth?

page 2

"…high accuracy for USV detection that" – What is the ground truth? Please mention here.

"…allows for the flexible use of any classification method." – What does that mean? /

"…USVs, outperforming previous tools" – Where is this shown? Where is the comparison on the same dataset?

…a highly accurate software to detect and.." – Mention that this is a supervised approach.

"…spectrogram is then analyzed in terms of its time-frequency plane,.." – What do you mean? The spectogram is already in the TF plan.

"…harmonic components (Figure 1D)" – To support reproducible research and open science, please provide the algorithm to the reader. In an open science approach provide the scripts (and data) to reproduce the tables and figures, where possible.

page 3

"…we manually inspected the spectrograms and labeled.." – Who is "we"? Experts on USVs? The authors? Did you listen to the resampled USVs? What is the inter-human detection rate? So what is the difference between experts? How did you select the test data?

"…the manual counting (4,441..)" – So a lot of FPs, right? Please report all the FPS and Fns.

"… artifacts of the segmentation process" – Where does this come from?

page 5

"…) to the sum over.." – Why the sum?

"…) to the sum over.." whole paragraph – Please represent your results in a figure or table! This is very tough to read and digest.…..

"Therefore, based on these results, we used the calculated inflection.." – Do I understand it right, that you criticize previous work because they required users to manually determine thresholds, but do the same for different parts of your model? For example with the inflection point.

"..In the test dataset," – How does the 14,5% FP rate compare to the one stated above?

"..In the test dataset," – which 7 USVS? out of 13 missing USVs?

"…compared the score for the label "noise"" – the linear regression between what?

"…suggesting high accuracy of VocalMat" – So about 4% of USVs are labeled as noise, right?

"…The rate of detected noise labeled as USV (false positive) was 5.60" – 42 out of 750 FPs, are falsely labeled as USVs, this probably should be mentioned in the abstract.

"Finally, the rate of USVs not detected (missed rate) was 0.28 Â{plus minus} 0.09% (mean Â{plus minus} SEM; median = 0.23; 95% CI [0.05, 0.51])." delete all this – "USVs not detected.." should not be discussed here. It is not related to classification problem and it is already discussed in section 2.2.

"identify approximately 1 in 75" – "identify approximately 1 in 71" – TNR and FPR are complementary so reporting one is enough. And 1 of 18 noises remains in the data.

"Characteristics of mislabeled USV candidates by VocalMat" – this paragraph does not give additional information, in my oppinion.

page 6

"(Table S1)." – Please add "in the appendix"

whole paragraph – Again a very ugly layout.. Why not put tables here, for example.

Why did you not try other pre-trained networks of DSQ for detection?

Your training data is highly imbalanced in classes U, C3, C and step down. Please discuss!

page 7

first paragraph – Again: Please do a graph or table!

"we manually inspected the spectrogram of the sample.." – Why? Do the same on the existing data set?

"We compared all four experimental conditions against each other and visually.." – Please compare to a SVM clustering.

" Since we did not train DeepSqueak with our.." – True, but your model also requires adaptation, so please argue why this is still a fair comparison.

"..when compared to DeepSqueak." – What about false positives?

"Detection of harmonic components" – Why did you report the detection rate for harmonics separately?

"..method to analyze the vocal repertoire of mice." – Which classes are grouped in a same class?

"The difference between the two groups was that in the treatment group,.." – Why did you not compare classification results? But used used diffusion maps instead?.

page 9

"…study that reported the sensitivity to.." – What about specificity?

"..This characteristic provides the possibility for unique.." – How feasible is it to use classifier for other vocal types?

page 10

"..a euclidean space.." – "..an Euclidean space.." – Is the caption of Figure 5 further explained in text?

page 11

(1) – give a reference, please! Which implementation did you use?

"Normalization and contrast enhancement" – Where significant parts cut out in the TF plane?

(3) – Add a reference, please.

"adjusted image" – What is considered one image? This means what regions was used to find L_in_ and H_ic_?

"We used a high pass filter (45 kHz) to eliminate sources of noise in the audible.." – Is not 45 kHz a very high threshold?

"..contrast adjustment.." – What is the adjusted image exactely?

".. If the value of the current pixel i" – What is t?

page 12

"..it was empirically chosen as t = 0.2 for our application. " – This contradicts the claim in the introduction, no?

"segments" – The super-windows?

rest of the paragraph – Please provide details how exactly?

"Local Median Filter" – Please define what a segment is!

First two sentence – What do you mean? Unclear! Where does the segmentation noise come from?

" of the pixels in a window that contains.." – Unclear, please provide details!

".file (now referred to as \Upsilon)." – What is this? Where is the starting point?Is it the whole segment? The frequency curve? A binary spectrogram?

"local parametric equations" – Why? Please motivate!

(9) What do you mean by that? What is the structure of \Upsilon'?

"..it was empirically chosen as t = 0.2 for our application. " – Why and how?

"The inflection point is then determined as the point with maximum.." – How is the maximum curvature calculated? This is again a manual set value, right?

Is tau calculated for each audio file separately? Why or why not?

page 13

"Our training dataset consisted of.." – This makes the classification task easier, right? Is this fair?

page 14

Table 1 – Please explain!

Table2 – This is clear, no?

"set (90%) and a validation.." – How? randomly?

" Diffusion maps for output visualization " – What do you mean here?

.."USVs as a distribution of probabilities.." – Really? This is explained in the following, right?

"..clustering of USVs.." – Explain (or put it later after definitions!) Visualization or clustering?

"Euclidean manifold" – Of which dimension? What do you mean? R^11^ is also Euliclidean!

"bandwith" – of what?

"..of the same label." – How? Maximum intra-cluster distance?

page 15

".."Euclidean space.." – What is this space? R^2^?

"s idea of moving from node to node" – How? Provide details?

(14) What is e_i_? e_j_? Please define!

"d through SVD decomposition" – "d through a SVD decomposition"

"Ms = D1/" – "Ms : = D1/" (to make clear this is a defintion.)

"..sake of notation, consider:" – "sake of notation, consider (for a unitary \Omega)"

(16), (17), (18) – This is not new, so give a reference, please.

"manifolds, we considered this a transfer learning problem" – Explain, please!

"L_s_" and "L_d_" – please give formal definitions.

Notation has to be clarified, keep fixed and precisely defined

page 16

(19) – In this setting n and d are constant, right? Please discuss!

"…n topology preservation (µ > 1)." – Please either use more or less details. Either put a reference and don't explain, or explain in more details.

"… Nf=∑i=1Ddi" – Why? Motivate!"common space F" – Which space? Is this defined?

"f the samples in order to predict the other half" – Here? Like above?

So the separation is not 90:10?

Is "n" defined?

page 17

"…and Prism 8.0 were used to…" – References, please!

"edited" – In how far? Please explain in full details, how you changed each figure!

"Data were first subjected" – Which data?

"Shapiro-Wilk" – Please add a reference!

"When homogeneity was a" – Was this assumed here or not? Please stat that preciesly.

"critical data collection and insights" – Why those lab members not co-authors?

page 18

"shows an interval of 10 ms." – Nice clarification, but formulate in text as precise formula, please

page 20

Can you please give a summary table of comparison with other approaches?

page 21

[4] [5] – format all references in the same way, please! E.g first names!

---

## [Author Response]

Reviewer #1:In the manuscript entitled "Analysis of ultrasonic vocalizations from mice using computer vision and machine learning", Fonseca at al. present a novel computational tool for analysis of mice ultrasonic vocalizations (USVs). This tool aims to (1) detect USV candidates from audio clips; (2) separate the USVs from the noise; (3) apply neural network technology for categorization of the various USVs to predetermined categories; They use this tool to analyze a large dataset of pup calls and validate their tool as well as compare it with other computational tools published in the last decade. Finally, they show how they can use diffusion maps and manifold alignment to distinguish between calls of distinct groups of pups.This tool is nice, but rather limited in its abilities and do not represent a conceptual or technical breakthrough. As for limitations, the tool presented here is designed, trained and validated for a specific type of murine calls (pup calls) and a predefined set of 11 categories, which may not cover all possible categories. As for technical advancement, the software combines criteria-based detection with neural network classification, similarly to previously published tools mentioned by the authors.

As pointed out by the reviewers and in the manuscript, there are several tools available for USV detection. These tools typically need substantial user inputs and customization to obtain high performance. Our goal was to develop a tool that can be broadly used by the community with minimal (in fact, no) need for user inputs. VocalMat outperforms all other tools, at least in the datasets that were made available by the developers. Thus, we believe VocalMat will largely benefit the community by providing a highly accurate software that can be easily used.

Moreover, although the authors claim superiority of their software over published tools, I wasn't convinced that this comparison was conducted in an unbiased and completely fair manner and that their tool really represents a significant advantage on other tools. For example, they used DeepSqueak on their dataset without properly training it for this type of data, while their tool is specifically trained for it. Moreover, the success rates the authors report for other tools here are significantly lower than those reported by the relevant publications, and do not fit comparisons made by others.

The reviewer raises an important concern. Indeed we have not trained DeepSqueak on our dataset. Doing so would take a reasonable amount of time given that DeepSqueak demands a substantial amount of manual interactions to refine their detection network, which includes manually rejecting or accepting detected calls and manually refining the threshold for power or another "score” used for detection. We did contact the developers of DeepSqueak to request additional audio recordings that they used to validate their tool. Unfortunately, the developers declined our request and maintained all datasets private precluding a more fair comparison among tools. Importantly, in their single audio recording made publicly available, VocalMat outperformed DeepSqueak. This is important because for this audio recording, DeepSqueak was trained on their dataset and showed lower performance than VocalMat that was trained in a completely different dataset. We have also expanded our comparisons with other tools by validating the dataset from USVSEG, which includes 15 audio recordings from mice in different experimental conditions. In these audio recordings, VocalMat also outperformed USVSEG in addition to identifying vocalizations that were often missed.

Also, I wasn't convinced that the software presented here will be working just as well as the authors claim on a distinct set of data coming from another lab and recorded in distinct conditions. My attempt to activate the software failed due to an error.

We understand the concerns of the reviewer. As we stated above, we have tried to obtain validated datasets from different laboratories but have failed with the exception of the developers of USVSEG. (It is relevant to emphasize that we made all our data and validation files publicly available since we deposited the manuscript on bioRxiv; unfortunately, most of the other tools do not make their datasets publicly available, making direct comparisons among tools difficult to perform). With the new datasets from USVSEG, we show that in audio files from different laboratories and in different experimental conditions, VocalMat still performs extremely well and outperforms other tools.

We sincerely apologize for the problems the reviewer faced when trying to use VocalMat. Shortly after our submission, we identified an URL check error in the installation procedure provided on our GitHub page and have updated it since.

Reviewer #2:Nice study overall, well-articulated and easy to follow. I also highly commend the authors for making all data + source code available.1. The goal of the study could be encompassing a wider aim. The authors state: "Our goal was to create a tool with high accuracy for USV detection that allows for the flexible use of any classification method." The task of USV detection is relatively simple (no surprise the authors get 98% accuracy), it is the accurate classification of the USV types that is of particular importance. I would suggest the authors rephrase to emphasize that aspect.

We thank that reviewer for the suggestion and we have now rephrased our study goal to encompass this wider aim.

2. "The output of the vocal classification provides the additional benefit of a probability distribution of vocal classes, allowing for the use of nonlinear dimensionality reduction techniques to analyze the vocal repertoire". Probably this needs some rephrasing into something like 'the output of the classifier enables the exploration of the probabilistic class membership'. More importantly though, that is not a pre-requisite for any dimensionality reduction techniques (linear or not). Dimensionality reduction could be directly applied to the extracted features, it is not dependent upon the classifier outputs.

We thank the reviewer for the suggestion and we have now re-written this section of the manuscript to address this point.

3. "A linear regression analysis between manually validated data from different audio files and the true positives of the CNN revealed an almost-perfect linearity.…" I would expect to see a simple confusion matrix here assessing whether each of the USVs was correctly detected, rather than quantifying the number of USVs within each phonation (in which case indeed the methodology attempted by the authors would be appropriate). I think it is far more useful to assess on a case by case basis the USVs, and potentially determine whether e.g. one or more of the raw files was challenging for any reason. The authors could provide multiple confusion matrices 11x11 e.g. as a Supplementary Excel file.

We thank the reviewer for the suggestion and we have now added Table 1 to the manuscript with the statistics for the performance of VocalMat in detecting USVs. Additionally, we also added individual confusion matrices as suggested by the reviewer as a supplementary material (Figure S2).

4. "In order to evaluate the performance of VocalMat in detecting USVs compared to other published tools, we analyzed the same test dataset…" The authors' tool has an unfair advantage in this case, in that their algorithm was trained on the data collected under identical conditions like the test data. Moreover, the test data contains USVs from the same mouse.

We thank the reviewer for raising this point and we have re-written section 2.4 in the results (Performance of VocalMat compared to other tools) in line of this comment.

More precisely, to account for performance bias towards the dataset used for training and evaluation, we have tested VocalMat in two datasets from other laboratories. First, we now provide comparisons between VocalMat and USVSEG in 15 audio recordings provided by USVSEG. We show that in these audio recordings, VocalMat still performs exceptionally well and outperforms USVSEG. Second, we use the single example audio recording provided by DeepSqueak to test the performance of VocalMat and show that VocalMat outperforms DeepSqueak. (Note: the DeepSqueak developers declined our request to provide additional audio recordings that were used to test their performance for our validation). Thus, VocalMat performs well in a variety of datasets collected at different laboratories and experimental conditions.

Finally, It is also important to note that the test data do not contain USVs from the same mouse (we have now clarified this point in the text). We used a variety of experimental conditions to create our training and testing datasets and we did not use the same audio recordings for both. Thus, even though the data were generated in the same laboratory, we tried to incorporate as much variability as possible in our datasets. We have also clarified these points in the methods.

5. "In summary, VocalMat is a new tool to detect and classify mouse USVs with superior sensitivity and accuracy while keeping all the relevant spectral features…" It is not clear if the authors mean superior to other studies in the literature that attempted USV classification. For example Vogel et al. (2019) [https://doi.org/10.1038/s41598-019-44221-3], reported very similar (85%) accuracy using more mainstream ML approaches than attempted in this study with CNNs.

We thank the reviewer for the comment. We have now edited this concluding sentence to avoid comparisons with other tools.

Nevertheless, we would like to clarify our choice of using CNNs. For the proper use of techniques such as Random Forest and SVM, it is important for the input data (extraction of features of the USVs) to be of very high quality. In Vogel et al. (2019), the authors used commercially available software (Avisoft) to extract features from the detected USVs. While Avisoft performs reasonably well in stable conditions with no noise, in our experience the performance of Avisoft deteriorates in noisy conditions and also needs intense parameter tuning when running several audio files. This is a laborious process that precludes high throughput analysis of audio recordings. In contrast to RF and SVM, CNNs can extract features from raw images, making the method more robust to noise. (In fact, we have originally tested RF and show lower performance compared to CNNs).

6. If I understand correctly the methodology the authors used uses a single split of the data into training and testing, and data from a mouse could end up in both; thus the authors do not necessarily prove that their methods generalize well in new datasets. I would welcome the use of e.g. leave-one-mouse out assessment, and also the use of an additional external dataset collected under perhaps slightly different lab conditions (different lab) to see how well findings generalize. CNNs are extremely sensitive, and theoretical work has shown that adding imperceptible (visually) noise in images results in complete different results.

We thank the reviewer again for raising this important point and we have clarified these points in the text. Importantly, the training and test datasets were two independent groups of audio recordings. Thus, the data from one mouse was only used in one of the datasets. There was no overlap between the datasets. We have also tested VocalMat in audio recordings used by USVSEG (15 recordings) and DeepSqueak (1 recording) and show high accuracy of VocalMat in detecting USVs.

Reviewer #3:The technical details should be better explained, with formulas and / or algorithm descriptions in one piece (not separated in main part and methods).– Could you do a sensitivity analysis on their model for different number of observations?

We thank the reviewer for the suggestion and we have now incorporated more performance tests in the text, including a new Table 1 that provides statistics of performance in different audio recordings with different number of observations.

– What is the efficiency of the method?

As stated above, we now provide several measures of software performance for VocalMat on Table 1 using seven independent audio recordings demonstrating an accuracy between 97.84-99.24%. We have also tested audio recordings from other groups and we now provide a statistical description of the performance of VocalMat in the text (section 2.4).

– Some state-of the art comparisons are missing: Did you compare it to the Automatic Mouse Ultrasound Detector? To DeepSqueak for classification? (This was only used for detection right?)

We thank the reviewer for raising this important point. We did not compare our detection performance with A-MUD as other tools have shown better performance and usability. Thus, we decided to compare VocalMat to USVSEG and DeepSqueak. These two softwares also had audio files that were made publicly available (USVSEG provided 15 audio recordings and DeepSqueak provided 1 audio recording – the developers of DeepSqueak declined our request to provide more of their validation files). We now provide the details of these comparisons on section 2.4. We also used these two softwares in addition to MUPET and Ax to test their performance on our fully curated test dataset (also in section 2.4). It is important to emphasize two points. First, our dataset is the first fully curated: different investigators manually inspected the entire spectrogram and labeled each USV, allowing the quantification of missed vocalizations (other softwares do not appear to do the same). Second, since the original submission of this manuscript, we made all our files and validation publicly available (with exception of USVSEG, other tools do not make their datasets and validation files publicly available and DeepSqueek even declined our request for access to additional validation files preventing further comparative analysis). In terms of classification, DeepSqueak’s classification network is trained on only 5 USV categories, which would not allow a direct comparison between the tools regarding their classification performance.

– What are the number of mice and number of records per mouse? How did you get them to vocalize?

We thank the reviewer for asking these questions and we have now clarified the number of animals in the methods (each mouse was only recorded once) and also the origins of the audio recordings. We have tested young mouse pups that vocalize when separated from the home nest and also tested audio recordings from adult mice that vocalize when in social context.

– You have only used recordings of pups (5-15 d old). Do the results apply to adult mice?

We have now provided validation of VocalMat performance on a set of adult recordings from a different laboratory (USVSEG dataset) and show excellent performance of VocalMat without the need of any parameter change.

– You used recordings of 5 strains of lab mice. Did you test or control for strain differences?

Our goal was to use several strains of mice and several experimental conditions to maximize variability of data for the development of a more powerful tool. We have now clarified this point in the text. We have not tested or controlled for strain differences.

– There are many classifications that have been proposed, and so what was the basis for your syllable type classification? Can you please explain / motivate that more? Can mice discriminate these syllable types?

We used the classification system generally used by experts in the field (Scattoni et al. (2008, 2011) and Grimsley et al. (2011)). Previous works using a similar definition of USV types show that mice modulate their call type use based on context (Warren et al. (2018) and Matsumoto et al. (2018)), which indicates that mice are capable of discriminating the different types. We have clarified this point in the text.

– How do you deal with the general machine learning problem that no insights are provided into the features that the model uses to classify syllable types ?

In the context of supervised learning, such as what our model uses, the features are implicitly defined by the user assigning the labels to the USVs that compose the training dataset. Thus, the machine learns recurrent patterns or features that link all the samples with similar labels. The correct classification of vocalizations that represent the stereotyped version of a given vocal class suggests the machine learned the appropriate features for the classification task.

Is this method useful for recordings with a noisy background? Is there any reason to suspect that it might not work?

This is a good point raised by the reviewer. We have clarified in the text that we used audio recordings in very different experimental conditions, including with high ambient noise (we provide these details now in the methods and main text). We have also tested datasets from different laboratories. In our experience, VocalMat performs exceptionally well in audio recordings with ample variability in environmental noise.

page1The comments on the Abstract"…detected more than >98% of the USVs" –What is the ground truth?page 2"…high accuracy for USV detection that" – What is the ground truth? Please mention here.

We have clarified these points by stating in the abstract and introduction that these were manually validated audio recordings. We have also clarified the ground-truth in the text and methods.

"…allows for the flexible use of any classification method." – What does that mean?

We have removed this sentence from the text to avoid confusion.

"…USVs, outperforming previous tools" – Where is this shown? Where is the comparison on the same dataset?

We thank the reviewer for raising this point. In response to this and other queries, we have now re-written section 2.4 in the results describing in more detail that comparison with other tools. We have used 3 different sets of audio recordings. Two sets were provided by the developers of different tools and one set was curated in our laboratory. We have clarified these points throughout the text.

…a highly accurate software to detect and.." – Mention that this is a supervised approach.

Thank you for your suggestion. We have clarified that VocalMat uses a supervised approach.

"…spectrogram is then analyzed in terms of its time-frequency plane,.." – What do you mean? The spectogram is already in the TF plan.

Thank you for the comment. We have corrected the sentence.

"…harmonic components (Figure 1D)" – To support reproducible research and open science, please provide the algorithm to the reader. In an open science approach provide the scripts (and data) to reproduce the tables and figures, where possible.

We agree with the reviewer and since our original deposit of the manuscript on bioRxiv we have made all our datasets, all our validation files, and all our codes available.

page 3"…we manually inspected the spectrograms and labeled.." – Who is "we"? Experts on USVs? The authors? Did you listen to the resampled USVs? What is the inter-human detection rate? So what is the difference between experts? How did you select the test data?

We thank the reviewer for these questions. We have now clarified in the text that our dataset was manually curated by two investigators trained to detect USVs in spectrograms and discrepancies were discussed with a third investigator. We made all our audio files and curated data publicly available. To the best of our knowledge this is the only dataset available with manually inspected spectrograms for the manual identification of each USV.

"…the manual counting (4,441.." – So a lot of FPs, right? Please report all the FPS and Fns).

We report all metrics of performance in sections 2.2 and 2.3 and provide a new Table 1 with summary of performance for each of the audio recordings tested.

"… artifacts of the segmentation process" – Where does this come from?

The artifacts originate from the sequences of operations performed on the image (of the spectrogram) in order to segment the objects of interest (USVs). Some of these operations may detect agglomeration of pixels as an object of interest, thus generating artifacts. The contrast filter step effectively eliminates these artifacts without eliminating real vocalizations that have low contrast.

page 5"…) to the sum over.." – Why the sum?

The sum over all the USV types corresponding to the complement of the probability of a USV candidate being noise. In other words, P(USV) = 1- P(noise), which is equal to the sum of the probability distribution over all the USV types.

"…) to the sum over.." whole paragraph – Please represent your results in a figure or table! This is very tough to read and digest.…..

Thank you for your suggestion and we have now included Table 1 with the detail of these analyses.

"Therefore, based on these results, we used the calculated inflection.." – Do I understand it right, that you criticize previous work because they required users to manually determine thresholds, but do the same for different parts of your model? For example with the inflection point.

We thank the reviewer for bringing this point to our attention as we realized we haven’t made it clear in the original submission. The inflection point is calculated automatically for each audio file. There are no human inputs on setting this parameter. We have clarified this point in the text.

"..In the test dataset," – How does the 14,5% FP rate compare to the one stated above?

The False Positive rate of 14.5% (750 of 5171 vocals) is not further discussed because it only refers to the contrast filter step and does not represent the final performance of VocalMat. We have updated the Results section and now include several metrics in Table 1, including the False Positives for each audio file in our test dataset.

"..In the test dataset," – which 7 USVS? out of 13 missing USVs?

We have re-written the Results section to clarify these and other points. We have also provide a new Table 1 with the summary of the performance of VocalMat to facilitate reading the manuscript and comparing the numbers.

"…compared the score for the label "noise"" – the linear regression between what?

There is no linear regression associated with this sentence. This sentence was just an introduction to the paragraph and the analysis that followed. We have re-written the text for clarity.

"…suggesting high accuracy of VocalMat" – So about 4% of USVs are labeled as noise, right?

We have clarified this point in the text. In fact, VocalMat labels real

USVs as noise in only 0.96% of the cases. When we combined all missed

USVs--including the ones not detected by VocalMat--then the rate is 1.64%.

"…The rate of detected noise labeled as USV (false positive) was 5.60" – 42 out of 750 FPs, are falsely labeled as USVs, this probably should be mentioned in the abstract.

For clarity and simplicity we report the accuracy in the abstract, which takes into account the number of false positives.

"Finally, the rate of USVs not detected (missed rate) was 0.28 Â{plus minus} 0.09% (mean Â{plus minus}SEM; median = 0.23; 95% CI [0.05, 0.51])." delete all this – "USVs not detected.." should not be discussed here. It is not related to classification problem and it is already discussed in section 2.2.

Thank you for your suggestion, we have deleted this sentence from this part of the manuscript.

"identify approximately 1 in 75" – "identify approximately 1 in 71" – TNR and FPR are complementary so reporting one is enough. And 1 of 18 noises remains in the data.

Thank you for the comment.

"Characteristics of mislabeled USV candidates by VocalMat" – this paragraph does not give additional information, in my opinion.

Thank you for the comment. We decided to keep the paragraph to provide more transparency of the data to the reader and also to suggest ways to improve the classification method by flagging USVs with lower probabilities.

page 6"Table S1)." – Please add "in the appendix"whole paragraph – Again a very ugly layout.. Why not put tables here, for example.

Thank you for your suggestion. We have updated the text to include the reference to the appendix. We also now provide a summary of these results in the form of a table (Table 2 in Section 2.5).

Why did you not try other pre-trained networks of DSQ for detection?

Thank you for the suggestion. We have now tested the performance of DSQ with their own audio file and provide data on the performance of DSQ network. From these tests, it appears that VocalMat performs better than DSQ in detecting USVs without the need for any tuning of the network/parameters.

Your training data is highly imbalanced in classes U, C3, C and step down. Please discuss!

This is a great point. At this moment, we choose to use naturally occurring USVs in the training set and some specific types of USVs occur rather infrequently, making it difficult to obtain enough samples for training the network in equal amounts.

page 7first paragraph – Again: Please do a graph or table!

Thank you for the suggestion. We have updated the Results section and added Table 2 with a summary of the results discussed in Section 2.5.

"we manually inspected the spectrogram of the sample.." – Why? Do the same on the existing data set?

Thank you for asking that question that we have clarified now in the text. Indeed we have manually validated all datasets and made them publicly and freely available to anyone who wants to use it to test their tools and software.

"We compared all four experimental conditions against each other and visually.." – Please compare to a SVM clustering.

Thank you for your suggestion, but it is not clear how SVM would contribute to the quantification of the similarity between the manifolds. Of course, we provide an example of how the data could be analyzed and many other tools can be used to compare the vocal repertoire of animals with existing tools and future developments.

" Since we did not train DeepSqueak with our.." – True, but your model also requires adaptation, so please argue why this is still a fair comparison.

This is another good point and we thank the reviewer for raising it.

In an attempt to address this point, we now show the performance of

VocalMat and DeepSqueak on the single audio recording provided by

DeepSqueak. Without any tuning of our model, VocalMat outperforms DeepSqueak. We would be glad to perform further tests with DeepSqueak’s dataset, but unfortunately the authors declined our request to share their validation dataset. Again, a policy that is contrary to us, as we made since the deposit of the preprint all our datasets and algorithms available.

"..when compared to DeepSqueak." – What about false positives?

Thank you for inquiring about it. We have re-written the Results section and have added Table 2 with a summary of the performance for each tool. We report the False Discovery Rate which takes into account false positives (FDR=FP/(FP+TP)).

"Detection of harmonic components" – Why did you report the detection rate for harmonics separately?

Most other tools do not detect/classify harmonic components and it is not common sense to describe them in the field. Thus we decided to report that as an extra feature of VocalMat, as even direct comparisons with other tools are not possible.

"..method to analyze the vocal repertoire of mice." – Which classes are grouped in a same class?

Classes are not grouped. The goal of this analysis is to show how much overlap between the classes exist regarding the probability distribution over the labels for each USV.

"The difference between the two groups was that in the treatment group,.." – Why did you not compare classification results? But used used diffusion maps instead?.

We compared the classification result in another report (Zimmer et al. (2019)). Here we used this large dataset as an example of how one could use the richness of the VocalMat’s output to analyze repertoire as probability distributions rather than just the assigned label by the classifier.

page 9"…study that reported the sensitivity to.." – What about specificity?

Thank you for your comment. We do report the specificity (true negative rate). However, the study cited in this sentence does not provide details about their specificity and we unfortunately cannot discuss this metric in this context.

"..This characteristic provides the possibility for unique.." – How feasible is it to use classifier for other vocal types?

Definitely feasible, as long as it is trained to do so. With the pre-trained network, any new vocal type will be classified in function of the 11 types used for training.

page 10"..a euclidean space.." – "..an Euclidean space.." – Is the caption of Figure 5 further explained in text?

Thank you for the comment. We will lead to editorial decision to use a or an in this case. We have consulted with a copyeditor who suggested to keep a as the word starts with a vowel that sounds like a consonant. We have also provided explanatory text in the manuscript as requested.

page 11(1) – give a reference, please! Which implementation did you use?"Normalization and contrast enhancement" – Where significant parts cut out in the TF plane?

We thank the reviewer for the comment. We have provided details about the method we used. We did not cut out any significant parts of the TF plane. We used a high pass filter at 45 kHz, as mice vocalize above this frequency.

(3) – Add a reference, please."adjusted image" – What is considered one image? This means what regions was used to find L_in_ and H_ic_?

The full spectrogram being processed is considered the one image. L_in_ and H_in_ are obtained from the full spectrogram. We have included a reference to the MATLAB function used at each step of our detection pipeline.

"We used a high pass filter (45 kHz) to eliminate sources of noise in the audible.." – Is not 45 kHz a very high threshold?

According to experimental observations (Grimsley et al. 2011 and Heckman et al. 2016, for example) mice emit USVs above 45kHz, typically above 50kHz.

"..contrast adjustment.." – What is the adjusted image exactly?

The adjusted image is the image following the contrast enhancement step.

".. If the value of the current pixel i" – What is t?

The intensity value of the pixel. We have updated the text.

page 12"..it was empirically chosen as t = 0.2 for our application. " – This contradicts the claim in the introduction, no?

Our claim of having an automated method does not eliminate the existence of parameters, which are fixed and require no user input. We have updated the Results section and extended the comparison with other tools.

We now include several audio recordings from different labs and show that VocalMat outperforms other tools when using their own datasets. This provides further evidence that this set parameter does not require manual user input, as intended.

"segments" – The super-windows?rest of the paragraph – Please provide details how exactly?"Local Median Filter" – Please define what a segment is!First two sentence – What do you mean? Unclear! Where does the segmentation noise come from?" of the pixels in a window that contains.." – Unclear, please provide details!

We have updated the text to clarify these points.

".file (now referred to as \Upsilon)." – What is this? Where is the starting point? Is it the whole segment? The frequency curve? A binary spectrogram?

Υ (\Upsilon) is the cumulative distribution function of the values of C_k_ (C_k_ is defined as the ratio of the intensity of the pixels belonging to a segment over the intensity of a window, or bounding box, around that segment).

"local parametric equations" – Why? Please motivate!

We describe in the text our goal of finding the inflection point of a function (the CDF of C_k_ in this case). To that end, we model that function using local parametric equations and ultimately obtain the inflection point.

(9) What do you mean by that? What is the structure of \Upsilon'?"..it was empirically chosen as t = 0.2 for our application. " – Why and how?"The inflection point is then determined as the point with maximum.." – How is the maximum curvature calculated? This is again a manual set value, right?Is tau calculated for each audio file separately? Why or why not?

Upsilon is the CDF of C_k_. The curvature is calculated as indicated in Equation (9) or (10). This is not a manually set value as indicated in the text.

page 13"Our training dataset consisted of.." – This makes the classification task easier, right? Is this fair?

Our classification method uses supervised learning, so having a training dataset becomes a requirement.

page 14Table 1 – Please explain!Table2 – This is clear, no?"set (90%) and a validation.." – How? randomly?" Diffusion maps for output visualization " – What do you mean here?.."USVs as a distribution of probabilities.." – Really? This is explained in the following, right?"..clustering of USVs.." – Explain (or put it later after definitions!) Visualization or clustering?"Euclidean manifold" – Of which dimension? What do you mean? R^11^ is also Euliclidean!"bandwith" – of what?"..of the same label." – How? Maximum intra-cluster distance?

We have updated the text to clarify these points.

page 15".."Euclidean space.." – What is this space? R^2^?"s idea of moving from node to node" – How? Provide details?(14) What is e_i_? e_j_? Please define!"d through SVD decomposition" – "d through a SVD decomposition""Ms = D1/" – "Ms: = D1/" (to make clear this is a defintion.)"..sake of notation, consider:" – "sake of notation, consider (for a unitary \Omega)"(16), (17), (18) – This is not new, so give a reference, please."manifolds, we considered this a transfer learning problem" – Explain, please!"L_s_" and "L_d_" – please give formal definitions.Notation has to be clarified, keep fixed and precisely defined

We have updated the text to clarify these points.

page 16(19) – In this setting n and d are constant, right? Please discuss!"…n topology preservation (µ > 1)." – Please either use more or less details. Either put a reference and don't explain, or explain in more details."… Nf=∑i=1Ddi" – Why? Motivate!"common space F" – Which space? Is this defined?"f the samples in order to predict the other half" – Here? Like above?So the separation is not 90:10?Is "n" defined?

We have updated the text to clarify these points.

page 17"…and Prism 8.0 were used to…" – References, please!"edited" – In how far? Please explain in full details, how you changed each figure!"Data were first subjected" – Which data?"Shapiro-Wilk" – Please add a reference!"When homogeneity was a" – Was this assumed here or not? Please stat that preciesly."critical data collection and insights" – Why those lab members not co-authors?

Most of these points are not common practice in biological journals such as *eLife* and we leave at the discretion of the editors to require more information or references for these basic operations.

page 18"shows an interval of 10 ms." – Nice clarification, but formulate in text as precise formula, pleasepage 20Can you please give a summary table of comparison with other approaches?page 21[4] [5] – format all references in the same way, please! E.g first names!

We have updated the text to clarify these points.